# Golgi self-correction generates bioequivalent glycans to preserve cellular homeostasis

**Haik Mkhikian[1], Christie-Lynn Mortales[1], Raymond W Zhou[2], Khachik Khachikyan[1], Gang Wu[3], Stuart M Haslam[3], Patil Kavarian[1], Anne Dell[3], Michael Demetriou[1,2]***

[1]Department of Microbiology and Molecular Genetics, University of California, Irvine, United States; [2]Department of Neurology and Institute for Immunology, University of California, Irvine, United States; [3]Department of Life Sciences, Imperial College London, London, United Kingdom

**Abstract** Essential biological systems employ self-correcting mechanisms to maintain cellular homeostasis. Mammalian cell function is dynamically regulated by the interaction of cell surface galectins with branched N-glycans. Here we report that N-glycan branching deficiency triggers the Golgi to generate bioequivalent N-glycans that preserve galectin-glycoprotein interactions and cellular homeostasis. Galectins bind N-acetyllactosamine (LacNAc) units within N-glycans initiated from UDP-GlcNAc by the *medial*-Golgi branching enzymes as well as the *trans*-Golgi poly-LacNAc extension enzyme β1,3-N-acetylglucosaminyltransferase (B3GNT). Marginally reducing LacNAc content by limiting N-glycans to three branches results in T-cell hyperactivity and autoimmunity; yet further restricting branching does not produce a more hyperactive state. Rather, new poly-LacNAc extension by B3GNT maintains galectin binding and immune homeostasis. Poly-LacNAc extension is triggered by redistribution of unused UDP-GlcNAc from the *medial* to *trans*-Golgi via inter-cisternal tubules. These data demonstrate the functional equivalency of structurally dissimilar N-glycans and suggest a self-correcting feature of the Golgi that sustains cellular homeostasis.

*For correspondence: mdemetri@uci.edu

**Competing interests:** The authors declare that no competing interests exist.

## Introduction

Self-correcting mechanisms have evolved to maintain the integrity of critical biological pathways in the face of disruptive insults and stochastic uncertainty. Such mechanisms range from proofreading in DNA replication to functional redundancy of important cellular apparatuses. The galectin-glycoprotein lattice is a dynamic cell surface structure that globally regulates receptor localization and signaling (*Demetriou et al., 2001*; *Partridge et al., 2004*; *Lau et al., 2007*; *Dennis et al., 2009*; *Grigorian et al., 2009*). Its importance is underscored by its role in basic cellular processes such as signaling, apoptosis, endocytosis, differentiation, and cell growth, as well as its association with a wide range of diseases including immunity/autoimmunity (*Demetriou et al., 2001*; *Lee et al., 2007*; *Mkhikian et al., 2011*; *Li et al., 2013*; *Wang et al., 2015*; *Zhou, 2014*), cancer (*Dennis et al., 1987*; *Fernandes et al., 1991*; *Granovsky et al., 2000*; *Beheshti Zavareh et al., 2012*; *Croci et al., 2014*), and Type 2 diabetes (*Ohtsubo et al., 2005*; *Johswich et al., 2014*). However, a mechanism that homeostatically sustains the lattice is not known.

The lattice forms due to the multivalent interactions between extracellular galectins, a family of sugar binding proteins, and the disaccharide N-acetyllactosamine (LacNAc) present on Asn (N)-linked glycans attached to cell surface glycoproteins (*Hirabayashi et al., 2002*; *Brewer et al., 2002*; *Ahmad et al., 2004*). The vast majority of secreted and cell surface proteins are co- or post-

**eLife digest** Most proteins that are released from cells are modified with sugar molecules that allow the proteins to carry out their role properly. These modifications are called glycans, and are made from sugar subunits joined into chains or branched structures. Investigating how the structure of glycans is linked to their role is complicated by the fact that many different glycans exist, made up of different sugars and arranged into different structures.

Enzymes located in cell compartments known as the endoplasmic reticulum and the Golgi help to build the glycans. For example, the MGAT family of enzymes found in the Golgi generates branched glycans made up of sugar subunits called N-acetyllactosamine (LacNAc). These glycans form part of a molecular mesh on the surface of cells that controls how certain proteins embedded in the cell membrane behave. This is particularly important in immune cells: reducing the number of branches in the glycans weakens the mesh and causes the cells and their membrane proteins to behave inappropriately.

Mkhikian et al. have studied mice that lack specific MGAT enzymes, and so produce LacNAc glycans with drastically fewer branches than normal. Immune cells in these mice had glycans on their surface formed of LacNAc arranged in chains, rather than in short branched structures. These chains turned out to be biologically equivalent to branched LacNAc glycans, containing the same sugar subunits and allowing the immune cells to behave as normal. This suggests that the composition of glycans, rather than their structure, primarily determines their role.

Mkhikian et al. also found that the organization of the enzymes inside the Golgi is likely to be responsible for producing these equivalent glycans. A glycan is built up as it passes through the Golgi, with the branching enzymes located earlier in the Golgi than the extending enzymes. Therefore, if the branching enzymes fail to add LacNAc subunits to the glycan, the extending enzymes can step in later to add the missing components.

Overall, the results presented by Mkhikian et al. indicate that the large number of structurally diverse glycans may be reduced to a much smaller number of glycans with similar roles, based on subunit composition. This will simplify future studies on LacNAc glycans, and further work could focus on defining which other glycan structures share similar roles.

translationally modified by the addition of sugars in the ER. As these proteins transit through the ER and Golgi, their glycans undergo dramatic remodeling, generating a vast and heterogeneous array of glycoforms (*Kornfeld and Kornfeld, 1985*; *Schachter, 1991*). In the *medial* Golgi a group of enzymes, MGAT1, 2, 4, and 5, act to produce N-glycans with one, two, three, or four N-acetylglucosamine (GlcNAc) branches (*Schachter, 1986*). The subsequent addition of galactose by a family of galactosyl transferase enzymes produces the galectin substrate LacNAc (*Figure 1—figure supplement 1A*). The number of branches depends on the relative activity of the *medial* Golgi branching enzymes MGAT1, 2, 4, and 5 and the availability of their shared donor substrate UDP-GlcNAc (*Lau et al., 2007*; *Dennis et al., 2009*; *Grigorian et al., 2007*; *2011*). Alternating action of β1,3-N-acetylglucosaminyltransferase (B3GNT) and galactosyl transferase enzymes can generate a linear polymer of LacNAc (poly-LacNAc) at any given branch. Although the affinity of galectin binding to a LacNAc monomer is relatively weak, increased LacNAc valency through branching and poly-LacNAc extension can dramatically increase galectin avidity leading to a major impact on cell surface dynamics (*Hirabayashi et al., 2002*). In T cells for example, galectin - T cell receptor (TCR) interactions directly oppose ligand induced TCR clustering and signaling, thereby negatively regulating T cell development, antigen-dependent T cell growth, and autoimmunity risk.

Glycan analysis of tissues from glycosylation pathway deficient mice has revealed the presence of minor but unusual structures (*Stone et al., 2009*; *Takamatsu et al., 2010*; *Ismail et al., 2011*). The function of these changes is unclear, but some have suggested that the observed structural alterations may reflect production of bioequivalent glycans that are induced by communication between the cell surface and the Golgi (*Takamatsu et al., 2010*; *Dam and Brewer, 2010*; *Dennis and Brewer, 2013*). However, direct evidence supporting this possibility is lacking. Deficiency in the branching enzyme β1,6-N-acetylglucosaminyltransferase V (MGAT5) reduces avidity for galectin,

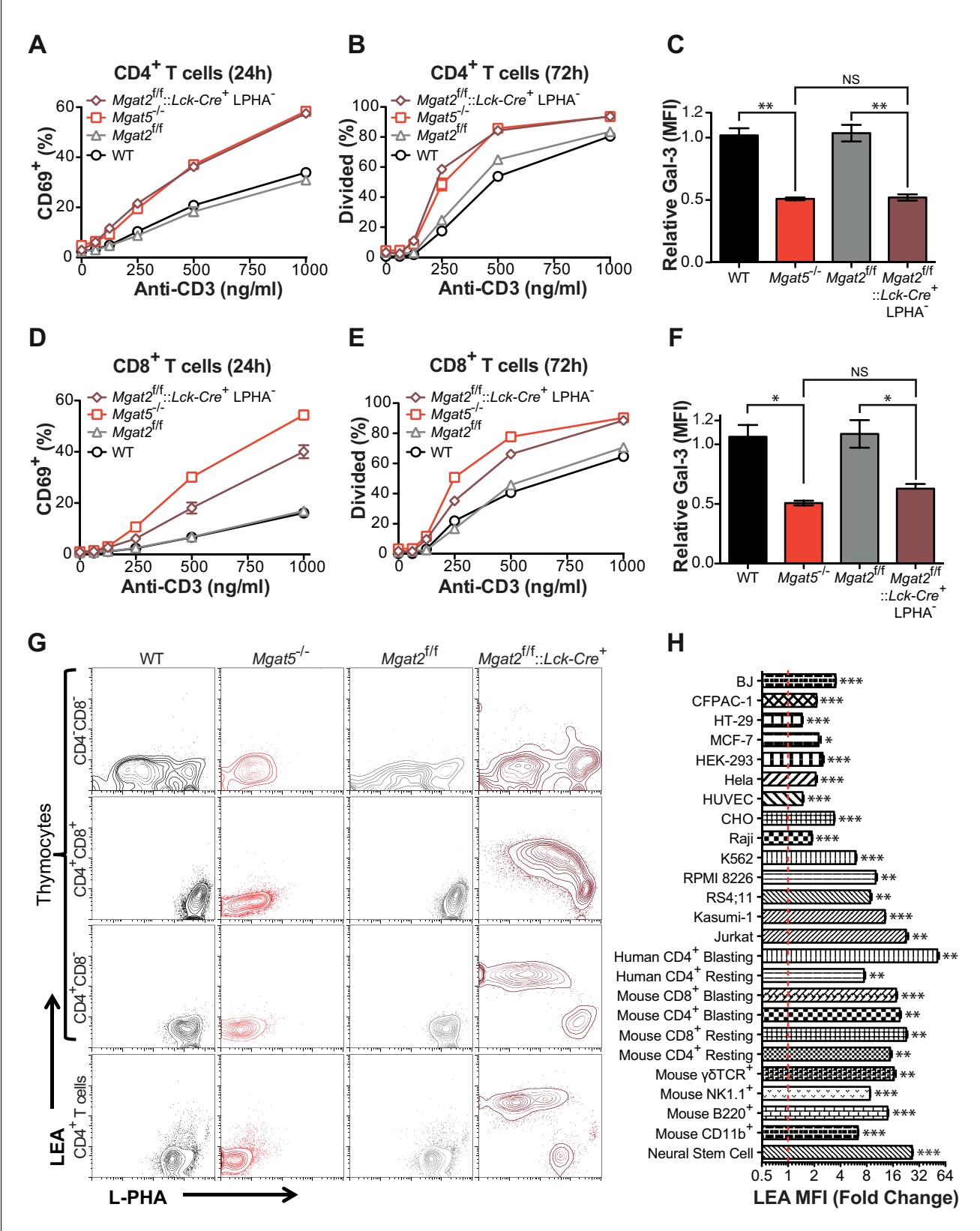

**Figure 1.** Compensation limits hyperactivity of *Mgat2* deficient T cells. (**A**, **B**, **D** and **E**) T cells were activated with plate bound anti-CD3 for 24 (**A** and **D**) or 72 (**B** and **E**) hours. CD4⁺ (**A** and **B**) or CD8⁺ (**D** and **E**) cells were analyzed for CD69 expression (**A** and **D**) or 5, 6-carboxyfluorescein diacetate

*Figure 1 continued on next page*

*Figure 1 continued*

succinimidyl ester (CFSE) dilution (**B** and **E**) by flow cytometry, gating on L-PHA⁻ cells where indicated. (**C** and **F**) T cells were analyzed for galectin-3 binding by flow cytometry, gating on CD4⁺ (**C**) or CD8⁺ (**F**) cells and L-PHA⁻ cells where indicated. Normalized geometric mean fluorescence intensity (MFI) is shown. Each mutant was normalized to its control. (**G**) Thymocytes and splenic T cells were analyzed for L-PHA and LEA binding by flow cytometry. (**H**) Cells were treated in culture with or without 500 nM SW for 72 hr followed by analysis of LEA binding by flow cytometry. Fold increase in LEA MFI of the SW treated sample compared to the untreated sample is presented. The red line marks one fold or no change. NS, not significant; *p<0.05; **p<0.01; ***p<0.001 (unpaired two-tailed t-test with Welch's (**C**, **F** and **H**) and Bonferroni correction (**C** and **F**)). Data show one experiment representative of at least three independent experiments. Error bars indicate mean ± s.e.m.

The following figure supplement is available for figure 1:

**Figure supplement 1.** The hexosamine and N-glycan biosynthetic pathways in mammals.

enhancing antigen dependent and independent TCR clustering/signaling, leading to development of spontaneous autoimmune disease (*Demetriou et al., 2001*; *Lee et al., 2007*). Based on the current model of the galectin-glycoprotein lattice, more severe reductions in branching should weaken the lattice further and result in greater T cell hyperactivity. Surprisingly, further limiting branching revealed that the Golgi apparatus has a remarkable capacity to buffer challenges to the strength of the galectin-glycoprotein lattice. Our analysis reveals a homeostatic mechanism built into the architecture of the Golgi apparatus that induces bioequivalent poly-LacNAc glycans that act to maintain the function of the galectin-glycoprotein lattice in the face of dysregulated Golgi branching.

## Results

### *Mgat2* deficiency does not increase T cell hyperactivity beyond *Mgat5* deficiency

To further investigate the role of branching in T cells, we generated T cell specific *Mgat2* deficient mice (*Mgat2*^f/f^::*Lck-Cre*⁺) (*Ye and Marth, 2004*). Loss of *Mgat2* is expected to limit N-glycans to a single branch, producing hybrid structures; although a second branch via MGAT4 activity is possible (*Figure 1—figure supplement 1A*). As the branching pathway declines in enzymatic efficiency going from MGAT1 to MGAT5, *Mgat2* deficiency also impacts a much greater percentage of cell surface glycans than *Mgat5* deletion (*Wang et al., 2001*). Examination of peripheral T cells from *Mgat2*^f/f^:: *Lck-Cre*⁺ mice indicated loss of *Mgat2* in most but not all T cells as assayed by flow cytometry with the plant lectin L-PHA (*Phaseolus vulgaris,* leukoagglutinin) (*Figure 1—figure supplement 1B*). β1,6GlcNAc-branched N-glycans produced by MGAT5 specifically bind L-PHA, structures that are also lost following *Mgat2* deletion (*Demetriou et al., 2001*; *Cummings and Kornfeld, 1982*). Surprisingly, *Mgat5* and *Mgat2* deficient CD4⁺ and CD8⁺ T cells displayed a similar degree of activation and proliferation in response to anti-CD3 (an antibody which induces TCR clustering and signaling) despite the more dramatic reduction in LacNAc branching in *Mgat2* deficient T cells (*Figure 1A,B,D and E*). This suggested that either the β1,6GlcNAc branch produced by the MGAT5 enzyme is uniquely important for regulating T cell activation or that a compensatory mechanism maintains galectin binding when the number of LacNAc branches is reduced. To evaluate for potential differences in total surface LacNAc content between *Mgat2* and *Mgat5* deficient T cells, galectin-3 binding at the cell surface was measured by flow cytometry. *Mgat5* deletion resulted in a significant reduction in the ability of CD4⁺ and CD8⁺ T cells to bind galectin-3 (*Figure 1C and F*), consistent with previously published results (*Demetriou et al., 2001*). However, *Mgat2* deficiency produced no additional decrease in galectin-3 binding (*Figure 1C and F*), suggesting comparable LacNAc content at the cell surface despite a marked reduction in LacNAc branches in *Mgat2* relative to *Mgat5* deficient T cells.

### Inhibition of LacNAc branching results in linear extension with poly-LacNAc

Since the branching pathway enzymes act sequentially, we hypothesized that compensatory maintenance of cell surface LacNAc content in *Mgat2* deficient T cells would primarily occur by poly-LacNAc extension of the MGAT1 generated branch (*Figure 1—figure supplement 1A*). To test this

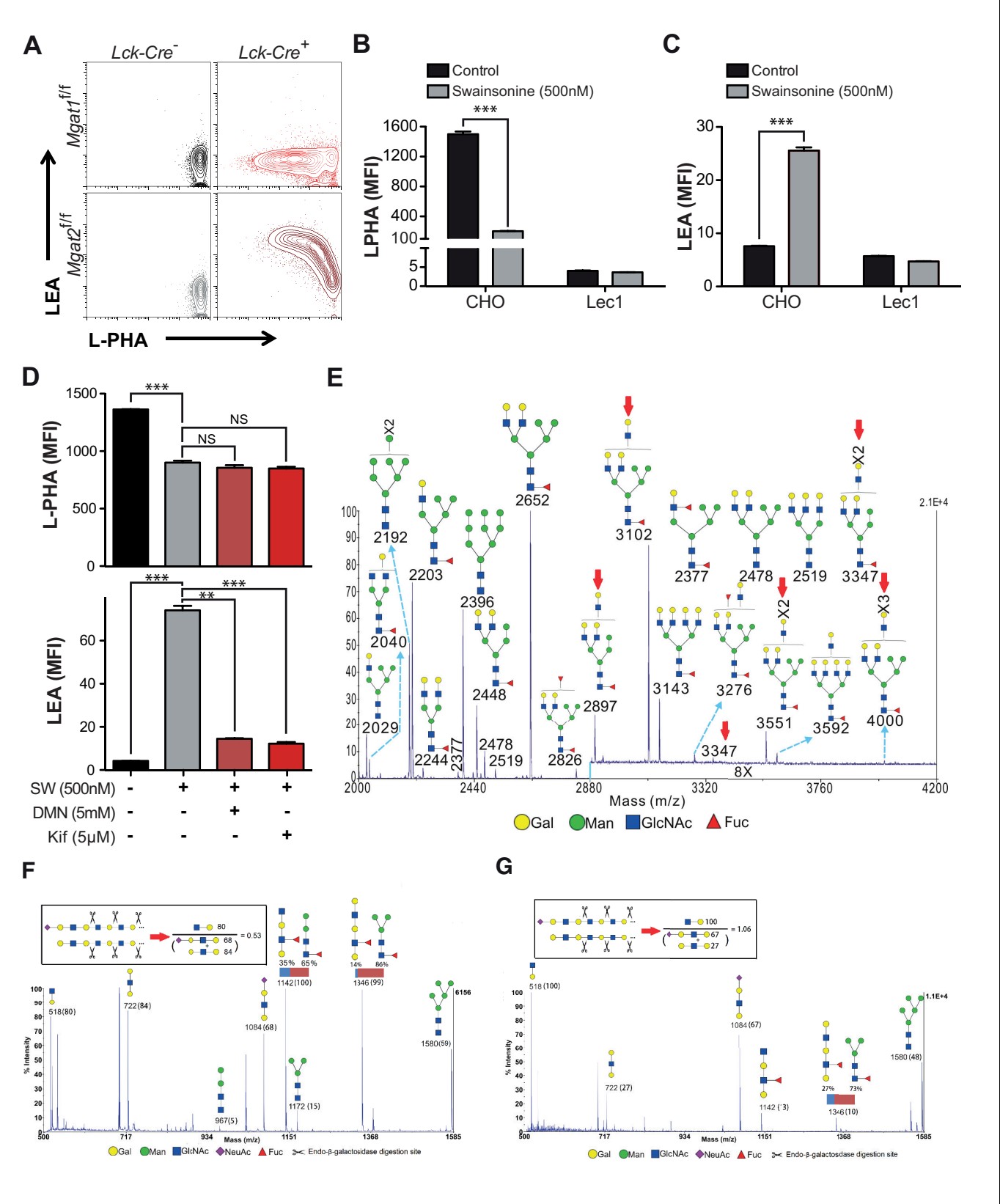

**Figure 2.** Branching deficiency induces poly-LacNAc on N-glycans. (**A**) Thymocytes were analyzed for L-PHA and LEA binding by flow cytometry, gating on CD4⁺CD8⁺ double positive cells. (**B** and **C**) CHO and Lec1 cells were grown in the presence or absence of 500 nM SW for 3 days followed by

*Figure 2 continued on next page*

*Figure 2 continued*

analysis for L-PHA (**B**) or LEA (**C**) binding by flow cytometry. (**D**) Resting primary human T cells were treated as indicated for 3 days and analyzed for L-PHA (upper) or LEA (lower) binding by flow cytometry, gating on live, non-blasting CD4$^+$ cells. (**E—G**) MALDI-TOF analysis of Sialidase A (**E**) or Endo-β-galactosidase treated N-glycans from Jurkat T cells treated without (**F**) or with SW (**E** and **G**). Hybrid glycans with extended antennae were observed in (**E**) at m/z 2897, 3102, 3347, 3551 and 4000, which are highlighted with red arrows. The % intensities of the peaks in (**F, G**) are shown in the brackets after m/z values. The signals at m/z 518, 722 and 1084 are derived from linear poly-LacNAc antennae and are used to represent the length of antennae. The calculation is shown above the spectrum. In addition, a minority of non-sialylated poly-LacNAc antennae were found to be internally fucosylated yielding GlcNAcβ1,3Galβ1,4(Fucα1,3)GlcNAcβ1,3Gal (m/z 1142) and Galβ1,4GlcNAcβ1,3Galβ1,4(Fucα1,3)GlcNAcβ1,3Gal (m/z 1346) upon digestion. MS/MS analysis of the peaks at m/z 1142 and 1346 also revealed the presence of isobaric pauci-mannose glycans, their relative abundances are indicated on the figure (see *Figure 2—figure supplement 3*). Ions are in the form of M$^+$Na$^+$. Peaks are annotated with putative structures according to the molecular weight, the glycan biosynthetic pathway and for (**E**), the MALDI-TOF analysis of the N-glycans before Sialidase A treatment (shown in *Figure 2—figure supplement 2*). NS, not significant; \*\*p<0.01; \*\*\*p<0.001 (unpaired two-tailed t-test with Welch's (**B–D**) and Bonferroni correction (**D**)). Data show one experiment representative of at least three independent experiments (**A–D**), except mass spectrometry (**E–G**), which was performed once. Error bars indicate mean ± s.e.m.

The following figure supplements are available for figure 2:

**Figure supplement 1.** Induction of poly-LacNAc structures occurs preferentially on N-glycans.

**Figure supplement 2.** MALDI-TOF analysis of N-glycans from SW treated Jurkat T cells MALDI-TOF analysis of N-glycans from SW treated Jurkat T cells.

**Figure supplement 3.** MS/MS analysis of endo-β-galactosidase digested glycans from Jurkat T cells.

prediction, T cells and thymocytes were stained with L-PHA as well as the *Lycopersicon Esculentum* lectin (LEA). LEA binds to poly-LacNAc structures containing at least three repeating LacNAc units (*Kawashima et al., 1990*; *Merkle and Cummings, 1987*). Wild type and *Mgat5*$^{-/-}$ T cells and thymocytes expressed very low levels of poly-LacNAc. However, in *Mgat2*$^{f/f}$::*Lck-Cre*$^+$ mice, loss of L-PHA staining was accompanied by a ~100 fold increase in LEA staining. The loss of L-PHA binding and the increase in LEA binding appeared to occur concurrently during the double positive stage of thymocyte development, shortly after the Lck promoter driven Cre is first expressed, and were maintained through the single positive stage and in peripheral T cells (*Figure 1G*). Treatment of various cell types with swainsonine (SW), a mannosidase II inhibitor which blocks N-glycan processing between MGAT1 and MGAT2 (*Figure 1—figure supplement 1A*), indicated that homeostatic upregulation of poly-LacNAc was a general feature of many cell types including epithelial, mesenchymal, and hematopoietic cells; with the greatest responses in the latter (*Figure 1H*).

Poly-LacNAc may occur on N-glycans as well as O-glycans and glycolipids (*Fukuda et al., 1986*; *Watanabe et al., 1979*). Furthermore, LEA has been reported to bind to high-mannose structures in addition to poly-LacNAc (*Oguri, 2005*). To investigate the structural basis for the increase in LEA staining, *Mgat2* deficient T cells were treated with PNGase F, an amidase which specifically cleaves N–glycans (*Maley et al., 1989*). PNGase F treatment of live cells incompletely removes N-glycans, with a four hour treatment of *Mgat2*$^{f/f}$ T cells reducing cell surface L-PHA binding by ~50% (*Figure 2—figure supplement 1A–B*). Nevertheless, the same treatment resulted in a >80% reduction in LEA binding in *Mgat2* deficient T cells, suggesting that the vast majority of LEA staining was due to cell surface N-glycans (*Figure 2—figure supplement 1C–D*). To further evaluate this question, we directly compared thymocytes derived from *Mgat2*$^{f/f}$::*Lck-Cre*$^+$ and *Mgat1*$^{f/f}$::*Lck-Cre*$^+$ mice (*Zhou, 2014*). *Mgat1* deficiency blocks all branching and poly-LacNAc extension in N-glycans, but not O-glycans or glycolipids. Indeed, unlike *Mgat2* deficiency, *Mgat1* deficient thymocytes do not show an increase in LEA staining concurrent with the loss in L-PHA staining (*Figure 2A*). Similarly, SW increases LEA staining in CHO cells but not *Mgat1* deficient CHO (Lec1) cells (*Figure 2B and C*). Furthermore, increased LEA staining induced by SW treatment of T cells was reversed by the mannosidase I inhibitors deoxymannojirimycin (DMN) and kifunensine (kif), which block the N-glycan pathway prior to MGAT1 (*Figure 2D*).

Poly-LacNAc content was also investigated using mass spectrometric glycomic methodologies. SW treated Jurkat cells, rather than T cells from *Mgat2*$^{f/f}$::*Lck-Cre*$^+$ mice, were used for this purpose to provide a sufficient amount of starting material for accurate analysis. MALDI-TOF-MS of N-glycans

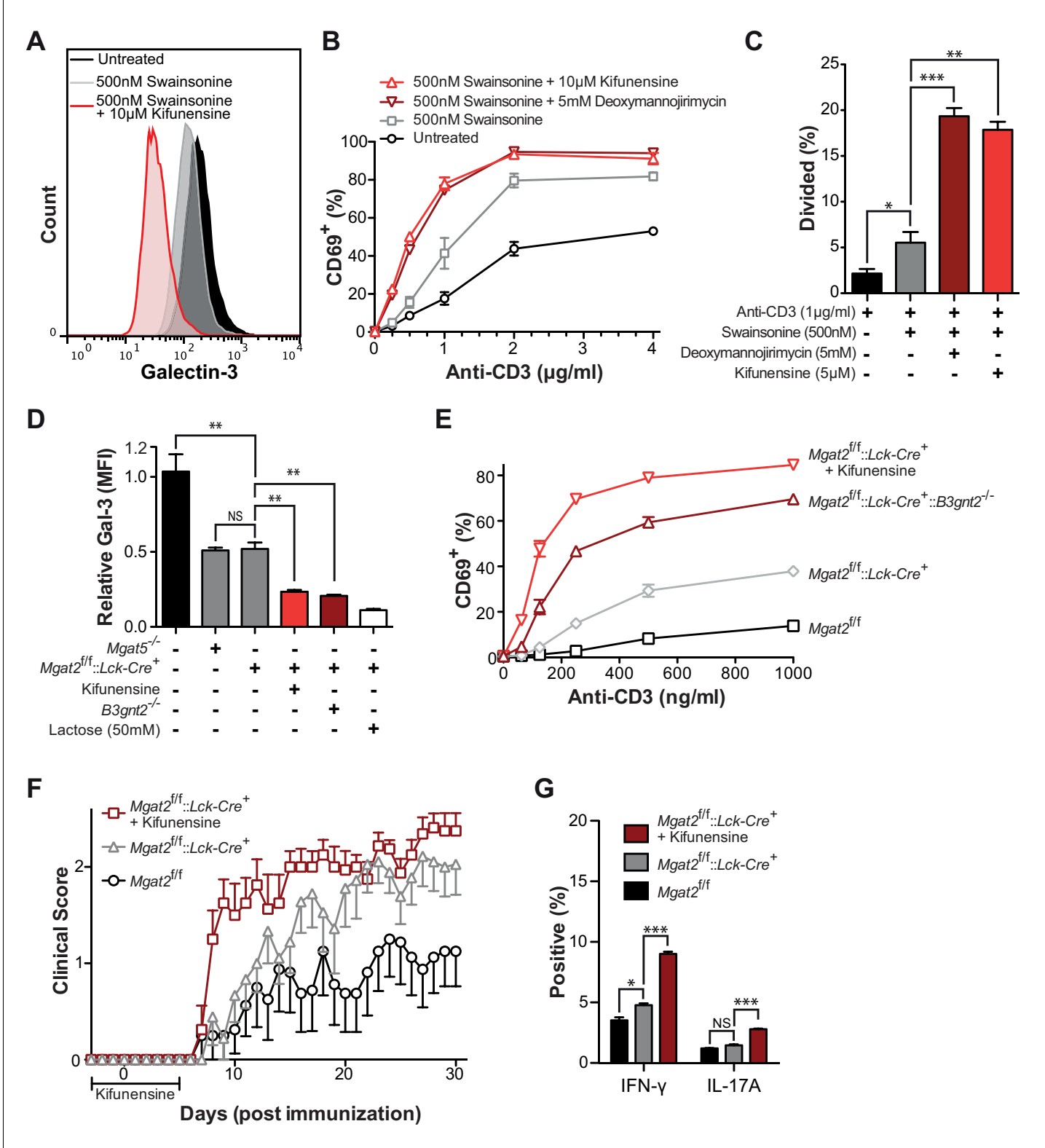

**Figure 3.** Poly-LacNAc compensation opposes T cell activation and autoimmunity. (**A**) Jurkat T cells were treated as indicated for 72 hr in culture and analyzed for galectin-3 binding by flow cytometry. (**B** and **C**) Human T cells were pre-treated as indicated for 72 hr in culture without stimulation, then activated with plate bound anti-CD3 for 24 (**B**) or 72 (**C**) hours and analyzed for CD69 expression (**B**) or CFSE dilution (**C**) by flow cytometry, gating on CD4[+] cells. (**D**) Mouse T cells were analyzed for galectin-3 binding by flow cytometry, gating on CD4[+] cells. Where indicated, mice were pre-treated for

*Figure 3 continued on next page*

*Figure 3 continued*

3 days with 0.2 mg/ml kifunensine in the drinking water. (**E**) Mouse T cells were activated for 24 hr with plate bound anti-CD3 and analyzed for CD69 expression by flow cytometry, gating on CD4$^+$ cells. Where indicated, mice were pre-treated for 3 days with 0.2 mg/ml kifunensine in the drinking water followed by 10 μM kifunensine during culture. (**F**) EAE was induced in age matched female C57BL/6 mice treated with or without kifunensine in the drinking water at 0.2 mg/ml from day -3 to 5, with day 0 indicating the time of immunization (n = 9 per group). (**G**) On day 30, splenocytes were isolated from representative mice of each EAE group and analyzed for cytokine expression by flow cytometry. NS, not significant; *p<0.05; **p<0.01; ***p<0.001 (unpaired two-tailed t-test with Welch's (**C**, **D** and **G**) and Bonferroni correction (**C**, **D** and **G**)). Data shown are one experiment representative of at least three independent experiments (**A–E**), except EAE (**F** and **G**), which was performed once. Error bars indicate mean ± s.e.m.

The following figure supplement is available for figure 3:

**Figure supplement 1.** Poly-LacNAc compensation opposes T cell activation and autoimmunity.

---

derived from SW treated Jurkat T cells confirmed the presence of hybrid N-glycans with up to two poly-LacNAc extended branches (m/z 2897, 3102, 3347, 3551, and 4000 in *Figure 2E*, *Figure 2—figure supplement 2*). This was qualitatively consistent with the presence of poly-LacNAc following SW treatment, but likely under-represents the length of poly-LacNAc extension due to the limited sensitivity of MALDI-TOF-MS for large poly-LacNAc structures. To better quantitate the change in poly-LacNAc, endo-β-galactosidase digestion was used. This enzyme cuts all internal Galβ1,4 linkages in poly-LacNAc structures, unless the GlcNAc on its reducing side is modified by fucose. Endo-β-galactosidase digestion produced GlcNAcβ1,3Gal (m/z 518) from internal linear poly-LacNAc antennae, together with Galβ1,4GlcNAcβ1,3Gal (m/z 722) and NeuAc-Galβ1,4GlcNAcβ1,3Gal (m/z 1084) from their non-reducing termini (*Figure 2F,G* and *Figure 2—figure supplement 3*). The ratio of the internal and terminal glycans examines the relative length of linear poly-LacNAc structures, with a higher ratio signifying longer chains. The non-treated cells had a ratio of 0.5, which increased to 1.1 in the SW treated cells (*Figure 2F and G*), confirming the presence of longer poly-LacNAc chains after SW treatment. *Mgat2* deficient T cells are expected to have a greater increase, given the ~100 fold increase in LEA binding compared to the ~20 fold increase in SW treated Jurkat cells. Together, these data indicate that severe LacNAc branching deficiency increases poly-LacNAc structures on N-glycans.

## Homeostatic poly-LacNAc opposes T cell hyperactivity and autoimmunity

To assess the functional consequences of poly-LacNAc up-regulation, we first reversed SW induced poly-LacNAc by blocking all branching using the mannosidase I inhibitors deoxymannojirimycin (DMN) or kifunensine. Whereas SW treatment alone moderately reduced galectin-3 binding, the addition of kifunensine dramatically reduced galectin-3 binding of Jurkat T cells (*Figure 3A*). As previously shown, SW treatment alone caused significant increases in both anti-CD3 induced activation and proliferation of primary human T cells (*Figure 3B,C* and *Figure 3—figure supplement 1*). However, the addition of kifunensine or DMN resulted in much greater hyperactivity, particularly at lower doses of anti-CD3. Careful titration of kifunensine in the presence of SW did not further reduce L-PHA binding, yet caused a dose dependent decrease in LEA binding and increase in CD69 induction, indicating that poly-LacNAc extension dose dependently regulates T cell activation thresholds (*Figure 3—figure supplement 1A–B*).

To further confirm the functional role of homeostatic induction of poly-LacNAc, mouse T cells deficient in both *Mgat2* and *B3gnt2* were generated. B3GNT2 is one of the major B3GNT enzymes responsible for poly-LacNAc branch extension in mouse T cells and poly-LacNAc up-regulation is expected to be limited in its absence (*Togayachi et al., 2007*). Indeed, T cells from *Mgat2*$^{f/f}$::*Lck-Cre*$^+$::*B3gnt2*$^{-/-}$ mice showed significantly reduced galectin-3 binding when compared to *Mgat2* deficient T cells, confirming reduced LacNAc content (*Figure 3D*). Directly comparing T cell activation between these lines showed that both genetic and pharmacological inhibition of homeostatic poly-LacNAc extension resulted in significantly increased T cell hyperactivity (*Figure 3E*).

Since the galectin lattice has been shown to inhibit autoimmunity, we next sought to determine the functional consequences of glycomic homeostasis in regulating the course and severity of experimental autoimmune encephalomyelitis (EAE), a model that mimics the autoimmune CNS pathology

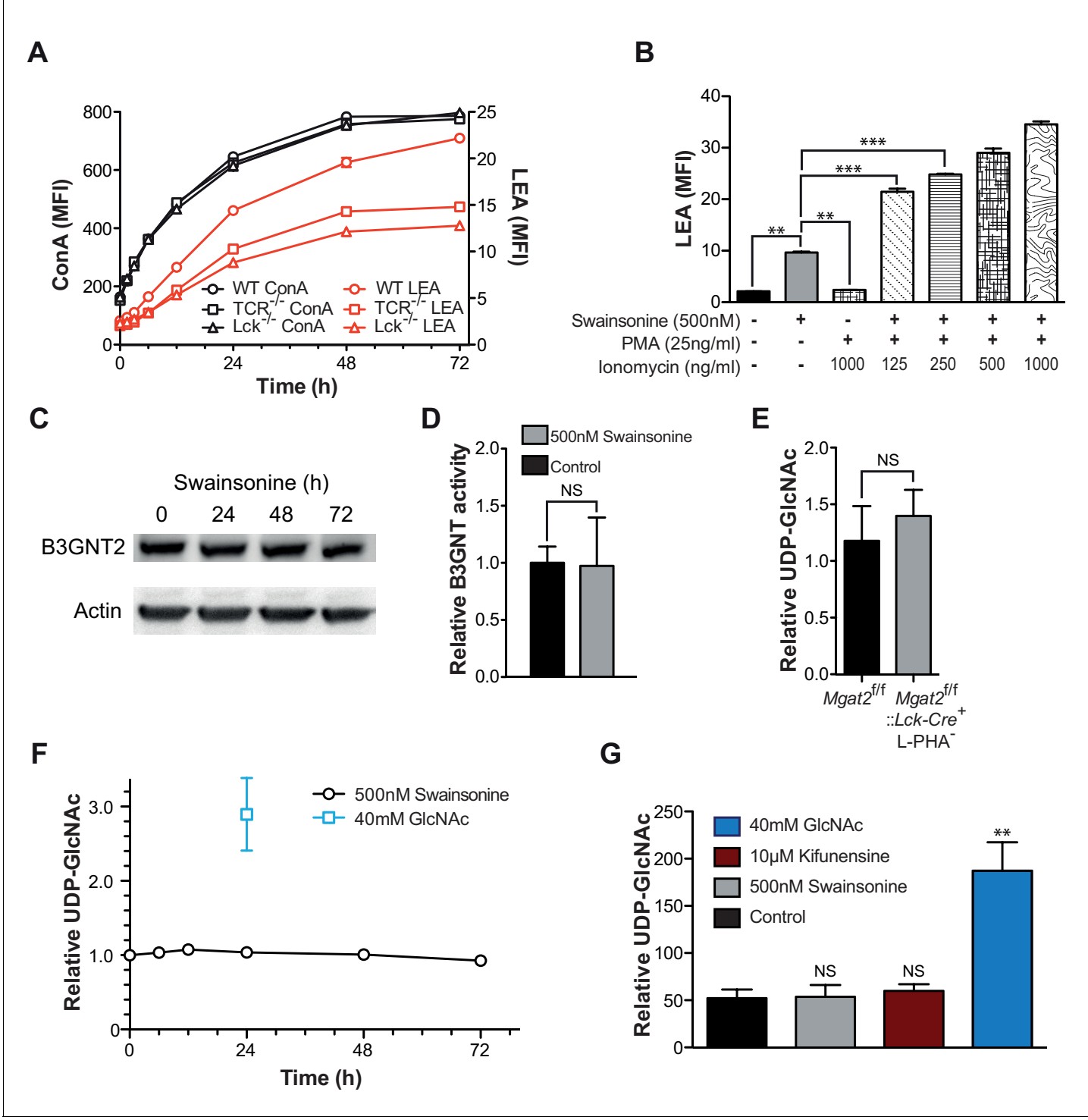

**Figure 4.** Increased TCR signaling and UDP-GlcNAc levels are neither necessary nor sufficient for poly-LacNAc induction. (**A**) WT, TCRβ[-/-] and Lck[-/-] Jurkat cells were treated with SW for the indicated times and analyzed for lectin binding by flow cytometry. (**B**) Jurkat cells were treated as indicated for 24 hr and analyzed for LEA binding by flow cytometry. (**C**) Lysates of Jurkat T cells treated with SW as indicated were immune-blotted with anti-B3GNT2 and actin. (**D**) B3GNT enzyme activity was measured in lysates of Jurkat T cells treated with and without SW for 72 hr. (**E–G**) Total cellular UDP-GlcNAc levels were measured in mouse T cells (**E**), or Jurkat T cells treated as indicated (**F**, **G**) via mass spectrometry (**E** and **G**) or a spectrophotometric method (**F**). Jurkat cells were treated for the indicated times (**F**) or for 24 hr (**G**). NS, not significant; *p<0.05; **p<0.01; ***p<0.001 (unpaired two-tailed t-test with Welch's (**B**, **D**, **E** and **G**) and Bonferroni correction (**B** and **G**)). Data show one experiment representative of at least three independent experiments. Error bars indicate mean ± s.e.m.

*Figure 4 continued on next page*

*Figure 4 continued*

The following figure supplement is available for figure 4:

**Figure supplement 1.** TCR signaling and cellular UDP-GlcNAc levels modulate the degree of poly-LacNAc induced by branching Deficiency.

of multiple sclerosis (*Grigorian et al., 2011*). *Mgat2*^f/f and *Mgat2*^f/f::*Lck-Cre*^+ mice were treated with either vehicle or kifunensine in the drinking water from day -3 to 5, with day 0 indicating time of immunization. As expected, vehicle treated *Mgat2*^f/f::*Lck-Cre*^+ mice displayed a significantly more severe EAE than vehicle treated *Mgat2*^f/f mice. However, kifunensine treatment of *Mgat2*^f/f::*Lck-Cre*^+ mice resulted in a dramatic increase in clinical score and disease incidence early in the disease course, a difference that narrowed later in the absence of kifunensine (*Figure 3F* and *Figure 3—supplement figure 1D*). Kifunensine treated mice also had more pro-inflammatory $T_H1$ and $T_H17$ cells compared to vehicle treated mice *in vivo* and following re-stimulation with MOG 35–55 *in vitro* (*Figure 3G*, *Figure 3—figure supplement 1E*). Taken together these data demonstrate a major role for homeostatic poly-LacNAc extension in controlling T cell growth, differentiation, and self-tolerance.

## Homeostatic poly-LacNAc is not induced by alterations in enzyme activity

As the galectin-glycoprotein lattice negatively regulates TCR signaling and TCR signaling promotes lattice strength (*Demetriou et al., 2001*; *Chen et al., 2009*), we hypothesized that LacNAc homeostasis may result from a feedback loop linking TCR signaling and cell surface LacNAc content. Such a mechanism, which depends on a cell surface sensor of Golgi activity/branching implies a temporal lag phase during which a defect is detected, a signal is sent, and then the Golgi generates the proper response. With this prediction in mind, Jurkat T cells were treated with SW for various times and analyzed for changes in cell surface glycosylation by L-PHA, LEA, and Concanavalin A (ConA), the latter a plant lectin that binds high-mannose structures increased by SW treatment (*Figure 4—figure supplement 1A–C*). Although the increase in LEA binding trailed slightly behind an increase in ConA binding, it began to increase within ~1.5 hr of SW treatment, indicating an almost immediate compensatory response. Loss of L-PHA staining exhibited the smallest slope, possibly reflecting the preferential cell surface retention of highly branched glycoproteins (*Figure 4—figure supplement 1C*).

Since the increase in LEA staining exhibited some delay, the role of TCR signaling in driving poly-LacNAc induction was assessed. Blocking TCR signaling genetically, with TCRβ and Lck deficient Jurkat lines, or pharmacologically, with MAP kinase inhibitors, only partially reduced the magnitude of the poly-LacNAc response induced by SW (*Figure 4A* and *Figure 4—figure supplement 1D*). More importantly, directly activating downstream TCR signaling with PMA and ionomycin did not induce poly-LacNAc up-regulation, although it further enhanced poly-LacNAc triggered by SW (*Figure 4B*). Thus, TCR signaling appears to be neither necessary nor sufficient for the homeostatic poly-LacNAc response, but does contribute to its magnitude in the context of deficient branching.

Since most surface receptors are glycosylated, there are a large number of possible additional cell surface sensors that may alter signaling to drive compensation. However, we reasoned that regardless of the upstream components, a mechanism that acts to increase poly-LacNAc production must do so by either increasing the activity of responsible enzymes or increasing substrate supply (UDP-GlcNAc or glycoprotein) for the reaction. It is unlikely that an increase in mono-antennary glycans could account for the observed compensatory response, as all branches are equally likely to be extended with poly-LacNAc (*Antonopoulos et al., 2012*; *Ishida et al., 2005*).

Microarray analysis of purified *Mgat2*^+/+ and *Mgat2*^-/- CD4^+ T cells revealed no significant changes in glycosylation genes known to impact poly-LacNAc production or UDP-GlcNAc biosynthesis (*Table 1*). Gene expression was altered in some genes unrelated to glycosylation, but these are likely related to downstream effects of *Mgat2* deficiency (*Table 2*). Comparing B3GNT2 protein levels in Jurkat T cells treated with or without SW also showed no difference (*Figure 4C*). Total poly-LacNAc enzyme activity in lysates from control and SW treated Jurkat T cells also revealed no

**Table 1.** List of N-glycan branching, poly-LacNAc production, and hexosamine pathway genes.

| Gene symbol | Gene description | p Value | Fold change |
|---|---|---|---|
| Man1a | mannosidase 1, alpha | 0.3602 | 1.0628 |
| Man1a2 | mannosidase, alpha, class 1A, member 2 | 0.4885 | -1.0551 |
| Man1b1 | mannosidase, alpha, class 1B, member 1 | 0.4397 | -1.0627 |
| Man1c1 | mannosidase, alpha, class 1C, member 1 | 0.4053 | 1.0748 |
| Man2a1 | mannosidase 2, alpha 1 | 0.8150 | -1.0168 |
| Man2a2 | mannosidase 2, alpha 2 | 0.3647 | -1.0787 |
| Man2b1 | mannosidase 2, alpha B1 | 0.0800 | -1.1350 |
| Man2b2 | mannosidase 2, alpha B2 | 0.6465 | 1.0331 |
| Man2c1 | mannosidase, alpha, class 2C, member 1 | 0.9826 | 1.0016 |
| Mgat1 | mannoside acetylglucosaminyltransferase 1 | 0.1005 | -1.1846 |
| **Mgat2** | **mannoside acetylglucosaminyltransferase 2** | **0.0000** | **-27.8780** |
| Mgat3 | mannoside acetylglucosaminyltransferase 3 | 0.1757 | 1.1377 |
| Mgat4a | mannoside acetylglucosaminyltransferase 4, isoenzyme A | 0.1486 | -1.2068 |
| Mgat4b | mannoside acetylglucosaminyltransferase 4, isoenzyme B | 0.2248 | 1.1345 |
| Mgat4c | mannosyl (alpha-1,3-)-glycoprotein beta-1,4-N-acetylglucosaminyltransferase, isozyme C (putative) | 0.5695 | 1.0539 |
| Mgat5 | mannoside acetylglucosaminyltransferase 5 | 0.4045 | 1.0666 |
| Mgat5b | mannoside acetylglucosaminyltransferase 5, isoenzyme B | 0.1450 | 1.1347 |
| B3gnt1 | UDP-GlcNAc:betaGal beta-1,3-N-acetylglucosaminyltransferase 1 | 0.5597 | -1.0587 |
| B3gnt2 | UDP-GlcNAc:betaGal beta-1,3-N-acetylglucosaminyltransferase 2 | 0.1503 | 1.1219 |
| B3gnt3 | UDP-GlcNAc:betaGal beta-1,3-N-acetylglucosaminyltransferase 3 | 0.4031 | 1.1072 |
| B3gnt4 | UDP-GlcNAc:betaGal beta-1,3-N-acetylglucosaminyltransferase 4 | 0.2923 | 1.1100 |
| B3gnt5 | UDP-GlcNAc:betaGal beta-1,3-N-acetylglucosaminyltransferase 5 | 0.1758 | 1.1742 |
| B3gnt6 | UDP-GlcNAc:betaGal beta-1,3-N-acetylglucosaminyltransferase 6 (core 3 synthase) | 0.5565 | 1.0511 |
| B3gnt7 | UDP-GlcNAc:betaGal beta-1,3-N-acetylglucosaminyltransferase 7 | 0.8028 | 1.0237 |
| B3gnt8 | UDP-GlcNAc:betaGal beta-1,3-N-acetylglucosaminyltransferase 8 | 0.6157 | 1.0443 |
| B3gnt9-ps | UDP-GlcNAc:betaGal beta-1,3-N-acetylglucosaminyltransferase 9, pseudogene | 0.2841 | 1.0856 |
| B3gntl1 | UDP-GlcNAc:betaGal beta-1,3-N-acetylglucosaminyltransferase-like 1 | 0.7979 | -1.0241 |
| B4galt1 | UDP-Gal:betaGlcNAc beta 1,4- galactosyltransferase, polypeptide 1 | 0.8003 | 1.0166 |
| B4galt2 | UDP-Gal:betaGlcNAc beta 1,4- galactosyltransferase, polypeptide 2 | 0.1023 | 1.1487 |
| B4galt3 | UDP-Gal:betaGlcNAc beta 1,4-galactosyltransferase, polypeptide 3 | 0.6938 | -1.0304 |
| B4galt4 | UDP-Gal:betaGlcNAc beta 1,4-galactosyltransferase, polypeptide 4 | 0.1378 | 1.1624 |
| B4galt5 | UDP-Gal:betaGlcNAc beta 1,4-galactosyltransferase, polypeptide 5 | 0.8380 | 1.0131 |
| B4galt6 | UDP-Gal:betaGlcNAc beta 1,4-galactosyltransferase, polypeptide 6 | 0.0645 | -1.2196 |
| B4galt7 | xylosylprotein beta1,4-galactosyltransferase, polypeptide 7 (galactosyltransferase I) | 0.4116 | -1.0764 |
| Gck | glucokinase | 0.0548 | 1.1856 |
| Hk1 | hexokinase 1 | 0.8593 | 1.0129 |
| Hk2 | hexokinase 2 | 0.4688 | 1.0771 |
| Hk3 | hexokinase 3 | 0.0236 | -1.2237 |
| Adpgk | ADP-dependent glucokinase | 0.4033 | -1.0787 |
| Gpi1 | glucose phosphate isomerase 1 | 0.7826 | -1.0173 |
| Gfpt1 | glutamine fructose-6-phosphate transaminase 1 | 0.2410 | -1.0941 |
| Gfpt2 | glutamine fructose-6-phosphate transaminase 2 | 0.0289 | 1.2116 |
| Gnpda1 | glucosamine-6-phosphate deaminase 1 | 0.6667 | -1.0364 |
| Gnpda2 | glucosamine-6-phosphate deaminase 2 | 0.9474 | 1.0054 |

*Table 1 continued on next page*

*Table 1 continued*

| Gene symbol | Gene description | p Value | Fold change |
| --- | --- | --- | --- |
| Gnpnat1 | glucosamine-phosphate N-acetyltransferase 1 | 0.3646 | -1.0839 |
| Nat9 | N-acetyltransferase 9 (GCN5-related, putative) | 0.2253 | 1.0995 |
| Pgm1 | phosphoglucomutase 1 | 0.7697 | -1.0243 |
| Pgm2 | phosphoglucomutase 2 | 0.7135 | -1.0360 |
| Pgm2l1 | phosphoglucomutase 2-like 1 | 0.8036 | -1.0167 |
| Pgm3 | phosphoglucomutase 3 | 0.6032 | -1.0529 |
| Pgm5 | phosphoglucomutase 5 | 0.2374 | 1.1326 |
| Uap1 | UDP-N-acetylglucosamine pyrophosphorylase 1 | 0.6555 | 1.0402 |
| Uap1l1 | UDP-N-acteylglucosamine pyrophosphorylase 1-like 1 | 0.2637 | -1.1092 |
| Nagk | N-acetylglucosamine kinase | 0.2907 | -1.1126 |
| Gale | galactose-4-epimerase, UDP | 0.5196 | 1.0579 |
| Gne | glucosamine (UDP-N-acetyl)-2-epimerase/N-acetylmannosamine kinase | 0.2860 | -1.0808 |

Gene symbol, gene description, *p* value and fold change of *Mgat2* deficient CD4[+] T cells relative to control are shown. The microarray confirms loss of *Mgat2* expression, highlighted in bold.

difference (*Figure 4D*). Thus, increased B3GNT enzyme activity is not responsible for the homeostatic up-regulation of poly-LacNAc induced by branching deficiency.

## Homeostatic poly-LacNAc is not induced by alterations in UDP-GlcNAc levels

Next we investigated whether increased cellular UDP-GlcNAc, the donor substrate for B3GNT enzymes, is triggered by severe branching deficiency. Measurement of total cellular UDP-GlcNAc levels by mass spectrometry and/or a colorimetric assay in cell lysates of purified *Mgat2*[+/+] versus *Mgat2*[-/-] T cells as well as Jurkat T cells treated with or without SW and kifunensine revealed no increase from branching deficiency (*Figure 4E–G*). Supplementing cells with GlcNAc raises cellular UDP-GlcNAc and branching (*Lau et al., 2007*; *Mkhikian et al., 2011*; *Grigorian et al., 2007*; *Grigorian et al., 2011*). In contrast, treatment of Jurkat T cells with GlcNAc did not increase LEA staining despite a 3–4 fold increase in cellular UDP-GlcNAc concentrations; although it further enhanced SW induced poly-LacNAc extension (*Figure 4F,G*, and *Figure 4—figure supplement 1E*). Over-expression of the three UDP-GlcNAc Golgi transporters (SLC35A3, SLC35B4, and SLC35D2) via transfection into Jurkat T cells also did not increase LEA staining (*Figure 5F*). Thus, much like TCR signaling, increased cellular UDP-GlcNAc levels are neither necessary nor sufficient to induce homeostatic up-regulation of poly-LacNAc. Not surprisingly, poly-LacNAc induction was blocked by 4-Fluoro-GlcNAc, a drug known to inhibit poly-LacNAc production by reducing UDP-GlcNAc biosynthesis (*Barthel et al., 2011*) (*Figure 4—figure supplement 1F–G*).

## UDP-GlcNAc redistribution from *Cis/Medial* to *Trans* Golgi triggers homeostatic poly-LacNAc

Glycosylation enzymes are organized in order of action from *cis* to *trans* along the secretory pathway. The branching enzymes (i.e. MGAT1, 2, 4 and 5) are localized to the *medial* Golgi, while the galactosyl transferase and B3GNT enzymes reside in the *trans* Golgi. The CMP-sialic acid transporter has a relatively restricted localization within the Golgi (*Zhao et al., 2006*), suggesting that sugar-nucleotide donors may be preferentially supplied to specific Golgi compartments. We hypothesized that UDP-GlcNAc supply and associated transporters may be restricted to the *cis/medial* Golgi compartment, thus preferentially driving branching over extension. Consistent with this hypothesis, raising total cellular UDP-GlcNAc increases branching by the *medial* Golgi enzymes (*Lau et al., 2007*; *Mkhikian et al., 2011*; *Grigorian et al., 2007*; *Grigorian et al., 2011*), but does not induce significant up-regulation of poly-LacNAc extension by *trans* Golgi B3GNT enzymes (*Figure 4F,G*, and *Figure 4—figure supplement 1E*). This suggests that cytosolic UDP-GlcNAc lacks direct access to the

**Table 2.** List of the top 50 differentially expressed genes.

| Gene symbol | Gene description | p Value | Fold change |
|---|---|---|---|
| Rpl36al | ribosomal protein L36A-like | 4.44E-16 | -15.6258 |
| Mgat2 | mannoside acetylglucosaminyltransferase 2 | 9.43E-14 | -27.8780 |
| Fn1 | fibronectin 1 | 5.83E-12 | -7.5845 |
| Cpm | carboxypeptidase M | 3.54E-10 | 3.1233 |
| Zbtb16 | zinc finger and BTB domain containing 16 | 5.45E-10 | -3.3239 |
| Mmp9 | matrix metallopeptidase 9 | 1.58E-09 | -3.9658 |
| Serpinb10-ps | serine (or cysteine) peptidase inhibitor, clade B (ovalbumin), member 10, pseudogene | 4.39E-09 | -5.0634 |
| Zfp69 | zinc finger protein 69 | 4.63E-09 | -2.1283 |
| Cd93 | CD93 antigen | 1.04E-08 | -2.3236 |
| Atp8a2 | ATPase, aminophospholipid transporter-like, class I, type 8A, member 2 | 2.64E-08 | -2.2711 |
| Gpr141 | G protein-coupled receptor 141 | 3.83E-08 | -3.4035 |
| Penk | preproenkephalin | 5.28E-08 | 3.8411 |
| Clec7a | C-type lectin domain family 7, member a | 8.21E-08 | -2.4336 |
| Lyz1 | lysozyme 1 | 1.06E-07 | -3.4287 |
| Mpeg1 | macrophage expressed gene 1 | 1.18E-07 | -4.3408 |
| Klra2 | killer cell lectin-like receptor, subfamily A, member 2 | 1.33E-07 | -4.4037 |
| Kit | kit oncogene | 2.44E-07 | -2.3150 |
| Plbd1 | phospholipase B domain containing 1 | 2.47E-07 | -2.9529 |
| Lyz2 | lysozyme 2 | 2.83E-07 | -3.6156 |
| Pld4 | phospholipase D family, member 4 | 3.00E-07 | -4.1375 |
| Ace | angiotensin I converting enzyme (peptidyl-dipeptidase A) 1 | 3.36E-07 | -3.3758 |
| Sirpa | signal-regulatory protein alpha | 3.40E-07 | -3.9975 |
| Clec12a | C-type lectin domain family 12, member a | 3.54E-07 | -4.4547 |
| Tlr13 | toll-like receptor 13 | 4.15E-07 | -6.5477 |
| Tnfrsf21 | tumor necrosis factor receptor superfamily, member 21 | 4.86E-07 | -3.4333 |
| Tgfbi | transforming growth factor, beta induced | 5.57E-07 | -3.6315 |
| Adrbk2 | adrenergic receptor kinase, beta 2 | 5.98E-07 | -2.0416 |
| Il12rb2 | interleukin 12 receptor, beta 2 | 7.32E-07 | -2.6927 |
| Csf1r | colony stimulating factor 1 receptor | 7.57E-07 | -5.9870 |
| Xcl1 | chemokine (C motif) ligand 1 | 9.55E-07 | -2.0969 |
| Mlh1 | mutL homolog 1 (E. coli) | 9.66E-07 | -2.0854 |
| Sulf2 | sulfatase 2 | 1.01E-06 | -2.9791 |
| Igsf6 | immunoglobulin superfamily, member 6 | 1.04E-06 | -3.0615 |
| Gpr56 | G protein-coupled receptor 56 | 1.10E-06 | -2.2352 |
| Fcna | ficolin A | 1.23E-06 | -4.7706 |
| Clec4a1 | C-type lectin domain family 4, member a1 | 1.29E-06 | -5.5202 |
| Gm11428 | predicted gene 11428 | 1.33E-06 | -2.5636 |
| Kcnj16 | potassium inwardly-rectifying channel, subfamily J, member 16 | 1.35E-06 | -2.5828 |
| Coro2a | coronin, actin binding protein 2A | 1.80E-06 | 1.5820 |
| Clec4a3 | C-type lectin domain family 4, member a13 | 1.88E-06 | -4.7329 |
| Muc13 | mucin 13, epithelial transmembrane | 1.91E-06 | -2.0524 |
| Abcd2 | ATP-binding cassette, sub-family D (ALD), member 2 | 2.49E-06 | -2.2046 |
| Cd68 | CD68 antigen | 2.56E-06 | -2.6229 |
| Cd34 | CD34 antigen | 2.65E-06 | -1.9429 |

*Table 2 continued on next page*

*Table 2 continued*

| Gene symbol | Gene description | p Value | Fold change |
| --- | --- | --- | --- |
| Cxcr6 | Sigmachemokine (C-X-C motif) receptor 6 | 2.68E-06 | -3.3244 |
| Mpo | myeloperoxidase | 2.80E-06 | -15.0742 |
| Car1 | carbonic anhydrase 1 | 2.95E-06 | -10.5252 |
| Pira11 | paired-Ig-like receptor A11 | 3.03E-06 | -3.3245 |
| Emr1 | EGF-like module containing, mucin-like, hormone receptor-like sequence 1 | 3.18E-06 | -6.1010 |
| Pira1 | paired-Ig-like receptor A1 | 3.23E-06 | -3.6538 |

Gene symbol, gene description, *p* value and fold change of *Mgat2* deficient CD4$^+$ T cells relative to control are shown. Data are ordered by *p* value.

*trans* Golgi compartment. To further evaluate this hypothesis, we examined the subcellular localization of the three UDP-GlcNAc Golgi transporters (SLC35A3, SLC35B4, and SLC35D2) and B3GNT2. Jurkat T cells were transfected with DDK or HA tagged versions of these proteins and examined for intra-Golgi localization using confocal microscopy-based line scan analysis. Intra-Golgi localization was determined by co-staining transfected cells with the *cis* and *trans* markers GM130 and TGN46, respectively, and determining the peak intensity of fluorescence along a line scan relative to these markers (*Dejgaard et al., 2007*). All three UDP-GlcNAc transporters localized to the *cis* and *medial* Golgi compartments while B3GNT2 localized to a later Golgi compartment (*Figure 5A–E*). In addition, overexpression of the UDP-GlcNAc transporters did not drive poly-LacNAc production (*Figure 5F*). These results suggest that the *trans* Golgi is comparatively deficient in UDP-GlcNAc, thereby limiting B3GNT activity and poly-LacNAc extension under steady state conditions.

As B3GNT requires UDP-GlcNAc to generate poly-LacNAc, we reasoned that these two factors must co-localize to drive homeostatic up-regulation of poly-LacNAc following disruption of branching activity. Co-localization may arise from movement of UDP-GlcNAc transporters to the *trans* Golgi and/or movement of B3GNT to the *medial* Golgi. However, SW treatment did not alter the Golgi localization of the three UDP-GlcNAc transporters or B3GNT2, arguing against this possibility (*Figure 5E* and *Figure 5—figure supplement 1A–D*). Alternatively, UDP-GlcNAc may directly shift from the *medial* to *trans* Golgi via the inter-cisternal transport system. Significant loss of activity of *medial* Golgi branching enzymes should acutely raise UDP-GlcNAc levels within the *medial* Golgi, with excess UDP-GlcNAc subsequently moving forward to the *trans* Golgi via inter-cisternal diffusion. Measuring UDP-GlcNAc by LC-MS/MS in a whole cell vesicular fraction isolated from post nuclear supernatant revealed significant levels of UDP-GlcNAc (*Figure 6—figure supplement 1A–B*). To ensure that UDP-GlcNAc was indeed within the vesicles as opposed to merely associated with the outer membrane, the post nuclear supernatant (PNS) was treated with either 0.1% Triton-X or 50 mM uridine monophosphate (UMP) for 15 min prior to isolation of the vesicular fraction by ultracentrifugation. Both treatments significantly reduced the amount of UDP-GlcNAc in the vesicular fraction, confirming UDP-GlcNAc was located in vesicles containing UDP-GlcNAc/UMP anti-porters (*Figure 6—figure supplement 1B–D*).

To evaluate the effects of reduced branching activity in the *medial* Golgi, we compared Jurkat T cells treated with and without kifunensine. Consistent with our hypothesis, UDP-GlcNAc levels were elevated in the vesicular but not the cytosolic fractions of kifunensine treated Jurkat T cells (*Figure 6A*). To examine the Golgi directly, we used antibodies to the *trans* marker TGN46 to immuno-isolate a Golgi enriched fraction from Jurkat T cell PNS. As expected from the physical connections between Golgi cisterna, western blotting for *cis, medial,* and *trans* Golgi markers confirmed that all three cisternal compartments were recovered with anti-TGN46 (*Figure 6B*). UDP-GlcNAc was elevated in the Golgi of kifunensine treated Jurkat T cells despite no difference in total cellular or cytosolic UDP-GlcNAc levels (*Figures 4G,6A,C*). Finally, we isolated a fraction enriched for *trans*-Golgi network (TGN) vesicles from Jurkat T cells by pretreating with Brefeldin A prior to pull down with anti-TGN46. Brefeldin A separates the *trans*-Golgi network from the rest of the Golgi by fusing the latter with the endoplasmic reticulum (*Martínez-Alonso et al., 2013*). TGN46 immuno-isolated vesicles from Brefeldin A treated Jurkat T cells were partially depleted for *cis* and *medial* Golgi markers relative to *trans* Golgi markers including B3GNT2, confirming relative enrichment of the

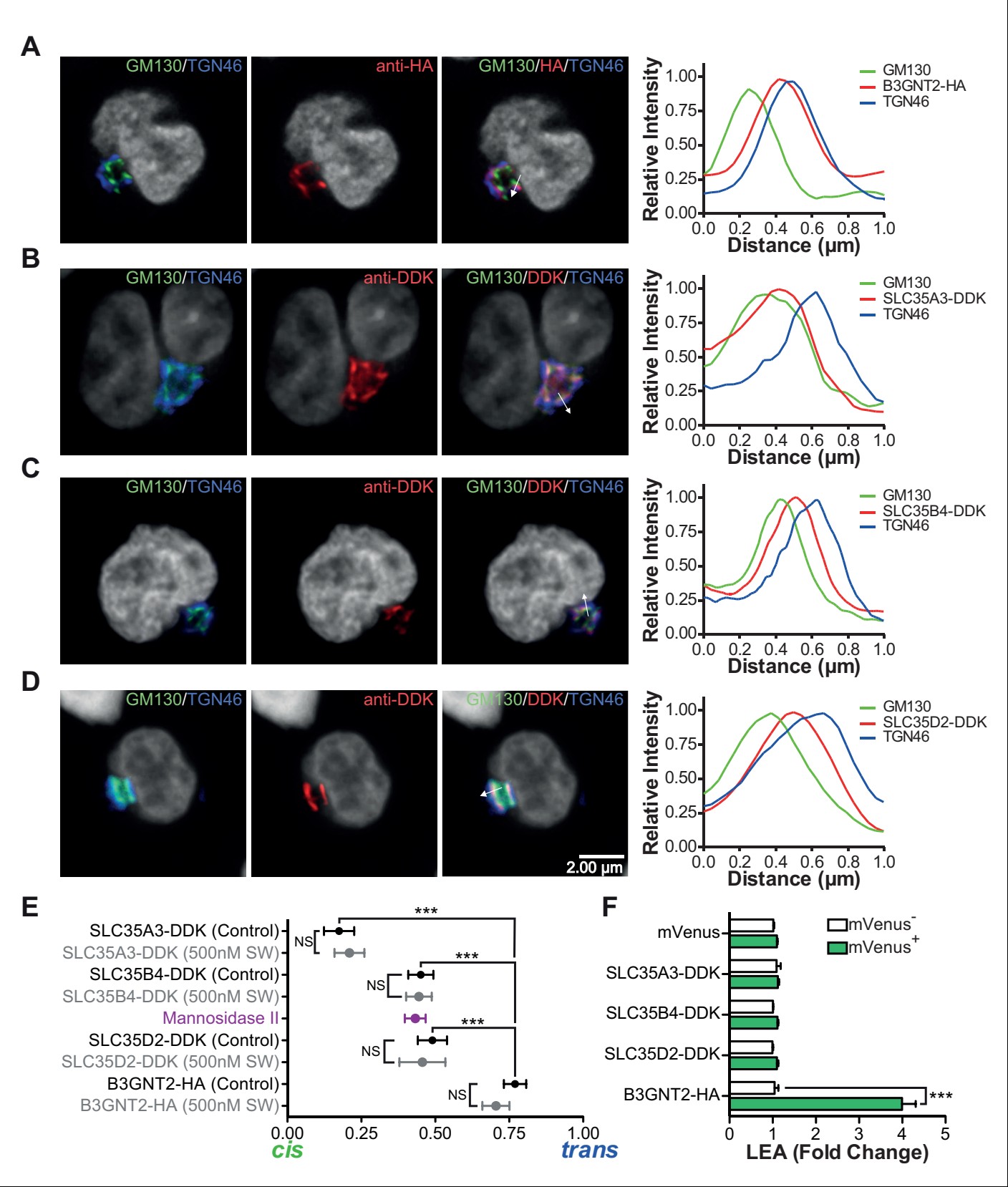

**Figure 5.** The UDP-GlcNAc transporters are localized to an earlier Golgi compartment than B3GNT2. (A–D) Jurkat T cells were transfected with plasmids containing B3GNT2-HA (A), SLC35A3-DDK (B), SLC35B4-DDK (C), or SLC35D2-DDK (D), cultured for 24 hr and then stained for GM130

*Figure 5 continued*

(green), TGN46 (blue) and either anti-HA (**A**) or anti-DDK (**B–D**) in red. Shown are representative confocal slices from transfected cells. The white arrows demonstrate areas deemed suitable for line scan analysis and the histograms (far right) demonstrate signal intensities along the line scans. DAPI staining is shown in grey scale. (**E**) Average localizations of the indicated Golgi proteins in Jurkat T cells treated with and without swainsonine (see *Figure 5—figure supplement 1*) relative to GM130 (*cis*) and TGN46 (*trans*) markers. Error bars show standard deviation. (**F**) Jurkat cells were transfected with mVenus alone or in combination with the four constructs of interest. After 48 hr, cells were analyzed for LEA binding by flow cytometry gating on mVenus⁻ and mVenus⁺ cells as indicated. NS, not significant; *p<0.05; **p<0.01; ***p<0.001 (unpaired two-tailed t-test with Welch's (**E** and **F**) and Bonferroni correction (**E**)). Data show images and histograms from representative cells (**A–D**), or pooled analysis from 30–40 cells per condition (**E**). Error bars indicate mean ± S.D. (**E**) or mean ± s.e.m (**F**).

The following figure supplement is available for figure 5:

**Figure supplement 1.** Branching deficiency does not cause relocalization of the UDP-GlcNAc transporters or B3GNT2.

*trans*/TGN compartment (*Figure 6D and E*). Vesicles from kifunensine treated Jurkat T cells demonstrated increased UDP-GlcNAc relative to control (*Figure 6F*), indicating that UDP-GlcNAc levels rise in the *trans* Golgi when *medial* Golgi branching activity is significantly diminished.

Although the mechanism for cargo transport between Golgi cisterna is incompletely understood, the prevailing model is cisternal maturation, where entire cisterna move forward with their cargo, while Golgi enzymes/transporters are pulled back to earlier cisterna by vesicular transport (*Glick and Luini, 2011*). However, this model has recently been expanded by data suggesting that small cargo transits the Golgi by diffusion via inter-cisternal tubules that vertically connect the *cis, medial,* and *trans* compartments (*Martínez-Alonso et al., 2013*; *Beznoussenko et al., 2014*; *Trucco et al., 2004*). Indeed, at least in CHO cells, there is evidence for functional continuity throughout sub-compartments of the Golgi (*Kim et al., 2001*). Golgi stacks, consisting of the *cis, medial,* and *trans* compartments, are also connected longitudinally to other Golgi stacks by tubules. Nocodazole, which interrupts longitudinal but not vertical Golgi tubules, had no effect on poly-LacNAc induction by SW in Jurkat T cells (*Figure 6G*). In contrast, pyrrophenone disrupts both longitudinal and inter-cisternal vertical tubules via inhibition of cytosolic phospholipase A$_2$-α (*San Pietro et al., 2009*). Pyrrophenone significantly blocked SW induced up-regulation of poly-LacNAc in Jurkat T cells in a dose dependent manner, as indicated by comparing its effects on LEA and ConA staining; the latter controlling for any changes in Golgi transport (*Figure 6H* and *Figure 6—figure supplement 1E*). In contrast, pyrrophenone had no effect on ConA, LEA, or L-PHA levels in control Jurkat T cells lacking SW (*Figure 6—supplement figure 1F–H*). We conclude that when branching enzymes under-utilize UDP-GlcNAc in the *medial* Golgi, UDP-GlcNAc accumulates and then shifts by diffusion to the *trans* Golgi via inter-cisternal tubules, thereby increasing substrate supply to B3GNT and triggering poly-LacNAc extension (*Figure 7*). However, our data do not exclude additional transfer of UDP-GlcNAc via vesicles or by cisternal maturation.

## Discussion

Our data demonstrates that structurally disparate glycans are functionally equivalent based on LacNAc content and suggests a novel self-correcting mechanism in the Golgi that sustains cellular homeostasis by ensuring LacNAc content is maintained at a minimal level within N-glycans. The complexity of glycan structures provides tremendous challenges for discerning the functional role of glycans in biology. Although N-glycan structures are highly diverse, redundancy of structural motifs within individual glycans has previously received little consideration as a means to collapse complexity and decipher glycan information. Deficiency of glycosylation pathway genes often give rise to new and unusual structures, triggering some to suggest that these may represent compensatory structures (*Stone et al., 2009*; *Takamatsu et al., 2010*; *Ismail et al., 2011*; *Dam and Brewer, 2010*; *Dennis and Brewer, 2013*). However, proof of functional equivalency has largely been lacking. The similar degree of hyperactivity between *Mgat5* and *Mgat2* deficient T cells indicates that poly-LacNAc extended hybrid structures are functionally equivalent to complex tri-antennary structures. Importantly, blockade of mannosidase II and *Mgat2* deficiency produced a similar phenotype, which was blocked/reversed by both mannosidase I inhibition and *Mgat1* deficiency, despite each targeting distinct biochemical steps and resulting in distinct structures. Thus, our data demonstrate that

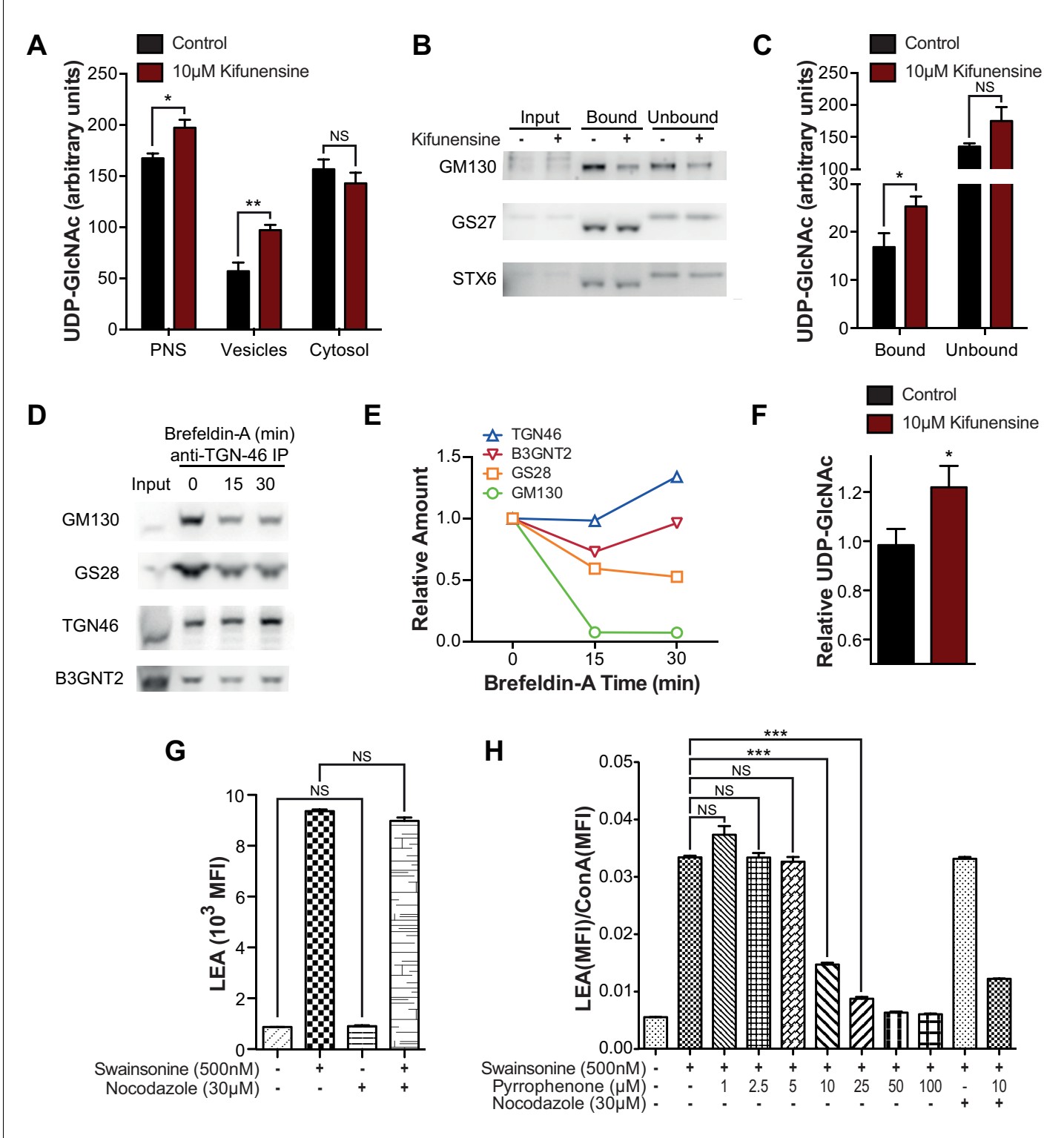

**Figure 6.** Intra-Golgi UDP-GlcNAc shifts to later Golgi compartments when use in the *medial* Golgi is inhibited. (A) LC-MS/MS quantitation of UDP-GlcNAc in post-nuclear supernatants (PNS), vesicular, and cytosolic fractions of kifunensine treated and untreated Jurkat T cells. (B) PNS (input) from kifunensine treated and untreated Jurkat T cells was used for Golgi enrichment via anti-TGN46 immuno-isolation followed by blotting for the Golgi compartment markers GM130 (*cis*), GS27 (*medial/trans*), and Syntaxin6 (*trans*). (C) LC-MS/MS quantitation of UDP-GlcNAc in anti-TGN46 bound and unbound fractions from kifunensine treated and untreated Jurkat T cell PNS. (D) PNS of Jurkat T cells treated for 0, 15, and 30 min with the Golgi disruptor Brefeldin A were used for anti-TGN46 immuno-isolation followed by blotting for B3GNT2 and the GM130 (*cis*), GS28 (*cis/medial*), and TGN46

*Figure 6 continued on next page*

*Figure 6 continued*

(*trans*) Golgi markers. (E) Quantitation of D, (F) LC-MS/MS quantitation of UDP-GlcNAc in anti-TGN46 immuno-isolates from Jurkat cells treated with Brefeldin A for 15 min. (G) LEA flow cytometric analysis of Jurkat T cells pre-treated with nocodazole where indicated for 45 min, followed by swainsonine where indicated for 5 hr. (H) Jurkat T cells that were treated with nocodazole +/- pyrrophenone as indicated for 45 min were treated with or without swainsonine for 5 hr and then analyzed for LEA and ConA binding by flow cytometry. Shown is the ratio of LEA MFI to ConA MFI for each condition. NS, not significant; *p<0.05; **p<0.01; ***p<0.001; (unpaired one-tailed (A, C, F and G) or two-tailed (H) t-test with Welch's correction and Bonferroni correction (H)). Data show one experiment representative of at least three independent experiments except F which shows combined data from two independent experiments. Error bars indicate mean ± s.e.m.

The following figure supplement is available for figure 6:

**Figure supplement 1.** Intra-Golgi UDP-GlcNAc shifts to later Golgi compartments when use in the *medial* Golgi is inhibited.

structurally diverse glycans are biologically equivalent if they share a similar amount of LacNAc. The overall glycan structure sets the total LacNAc content, thereby largely determining avidity of binding for galectins. It should be noted however, that although these differing glycoforms are clearly

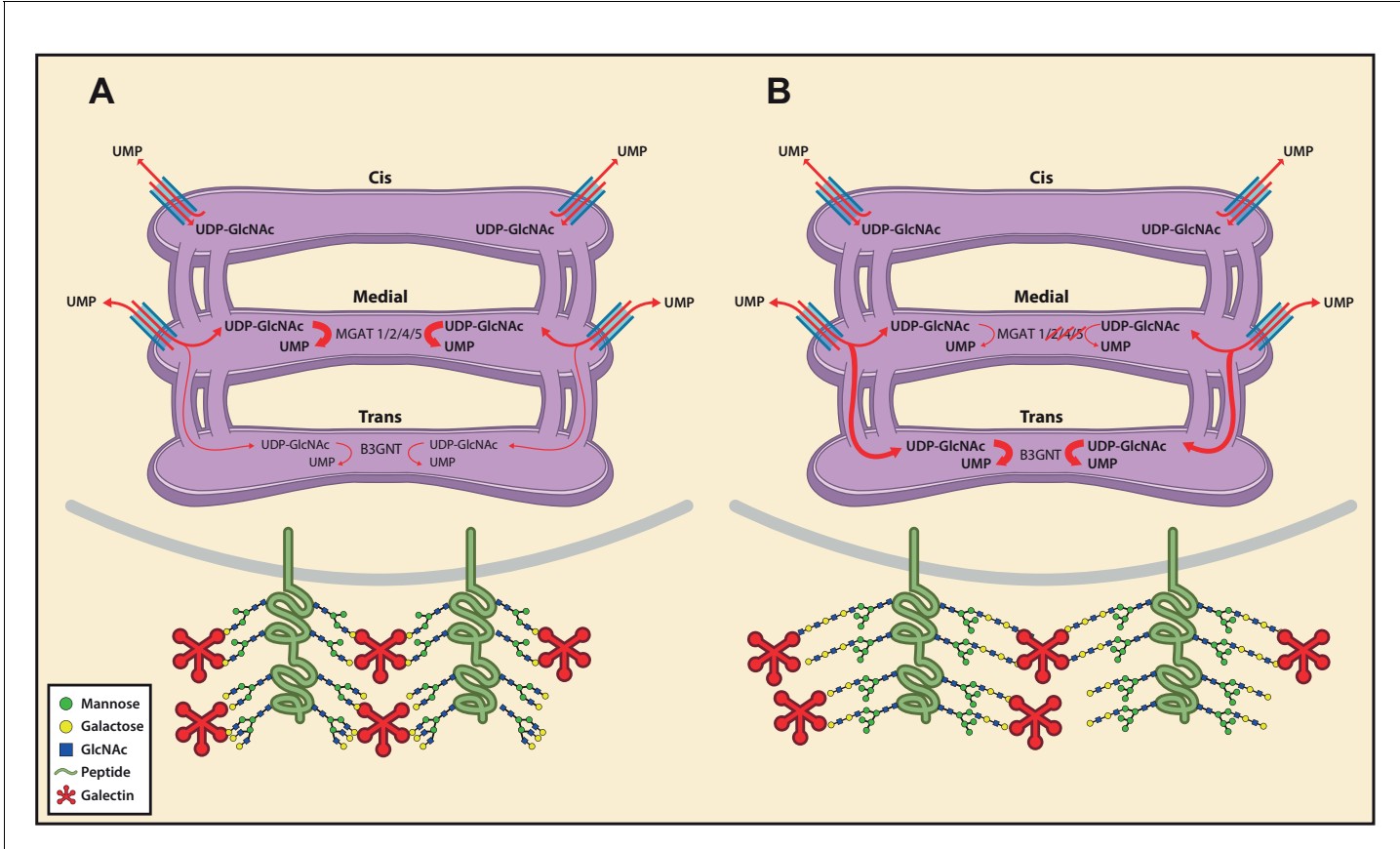

**Figure 7.** Model of an inherent Golgi self-correcting mechanism to maintain LacNAc homeostasis. The majority of UDP-GlcNAc entering the Golgi is supplied to its early compartments by the UDP-GlcNAc/UMP antiporters, which are preferentially localized to the *cis/medial* Golgi. Under branching proficient conditions (A), UDP-GlcNAc is used by the branching enzymes MGAT1, 2, 4, and 5, with little unused UDP-GlcNAc supplying the poly-LacNAc synthesizing B3GNT enzymes. The resulting array of N-glycans produced thus contains more LacNAc branches than linear LacNAc polymers (A). When the branching pathway is disrupted (B), or presumably the Golgi is otherwise stressed, leading to reduced UDP-GlcNAc usage in the *medial* Golgi, UDP-GlcNAc is driven forward (at least partially through intercisternal tubules) and promotes production of bioequivalent poly-LacNAc containing glycans by the *trans* Golgi-resident B3GNT family of enzymes. Under this scenario, loss of LacNAc branches is balanced by increased production of linear LacNAc polymers, a self-correcting ability that serves to maintain cell surface LacNAc density and thus the galectin-glycoprotein lattice (A). In the context of T cells, this homeostatic mechanism acts to curtail catastrophic T cell hyperactivity and promotes self-tolerance.

overlapping, they might not be strictly identical due to distinct geometries of LacNAc presentation or modification by fucose or sialic acid.

Our data support a model of N-glycosylation that emphasizes the significance of functional units within an N-glycan, such as LacNAc groups, rather than uniqueness of the overall glycan structure in determining biological function. The National Academy of Sciences position paper on Glycoscience has proposed a major focus on glycan structural determination (*National Academy of Sciences, 2012*). Our data suggests that glycan diversity may be collapsed into a much smaller group of bio-equivalent structures based on the number of structural subunits within the overall glycan that bind to a given animal lectin. Indeed, it may be possible that through further studies exploring bioequivalence among glycans, a glycan code can be deciphered based on interaction with animal lectins. Such an approach may greatly alleviate the complexity in determining structure-function relationships and relevance to human health.

Essential biological systems employ homeostatic mechanisms to maintain cellular integrity. N-glycosylation is an essential biosynthetic pathway necessary for cellular homeostasis and mammalian development (*Dennis et al., 2009*), yet lacks a known self-correcting mechanism. Our data suggest that homeostatic control of LacNAc content does not result from changes in enzyme levels/activity or metabolic production of UDP-GlcNAc substrate; but rather appears to arise from the structure of the Golgi. A defining feature of the Golgi is its polarized compartmentalization, with a *cis* to *trans* organization. Glycosylation enzymes are arranged along the Golgi roughly in the order in which they act. Why the Golgi has evolved this organization remains an open question. Our data indicate that the UDP-GlcNAc transporters preferentially supply UDP-GlcNAc to the *cis/medial* over *trans* Golgi, thus prioritizing branching over extension. Despite the presence of continuities between Golgi cisterna and previous work arguing that the Golgi is functionally interconnected (*Kim et al., 2001*), our data suggest that the segmented organization of enzymes and transporters allows for local depletion of substrate prior to diffusion to other compartments. In this manner, the evolutionary placement of the B3GNT poly-LacNAc extension enzymes in the *trans* Golgi provides a backup mechanism that captures unused UDP-GlcNAc from the *medial* Golgi shunted via inter-cisternal tubules. This backup system may also explain the lack of evolutionary pressure to produce genetic redundancy in the Golgi branching enzymes in mammals and the dramatically more severe phenotype seen with *Mgat1* deficiency, which blocks all production of N-glycan LacNAc, compared to other *Mgat* genes (*Ioffe and Stanley, 1994*; *Metzler et al., 1994*).

The Golgi is charged with maintaining the integrity of the glycome/lattice or risking disease. Thus, fidelity of glycan biosynthesis must be maintained under a variety of cellular states and be active virtually at all times. High protein transit rates present a significant challenge to the Golgi and have the potential to reduce the *medial* Golgi branching efficiency by decreasing the time available for branching reactions. Under this scenario, having a back-up system that provides poly-LacNAc extension in the *trans* Golgi by capturing unused UDP-GlcNAc from the *medial* Golgi would be critical for the maintenance of the lattice at a minimal essential level. Even in an unstressed system, a second step in the assembly line would act to counteract the moment to moment variability and stochastic uncertainty of glycan synthesis. Severe branching deficiency (induced by SW treatment or *Mgat2* deletion) uncovers this continual process. Such a mechanism is akin to DNA repair, which is occurring all of the time but made more apparent in the context of physiological or external stress.

A clear understanding of the network of interacting factors that coalesce to determine the state of the galectin-glycoprotein lattice is required for the successful exploitation of its therapeutic potential. Interventions must be able to overcome or at least take into account the homeostatic cellular response to the initial disturbance. For example, the effectiveness of SW as an anti-cancer therapeutic is likely limited by compensatory poly-LacNAc production (*Goss et al., 1994*; *1997*). Thus, therapeutic strategies aimed at disrupting the lattice should block branching and extension simultaneously, or target a more proximal regulator such as UDP-GlcNAc metabolism or transport. In light of these considerations, the lattice approach to cancer treatment may merit re-visitation. Conversely, a deeper knowledge of the central regulatory machinery that determines total LacNAc content will also be required for therapeutic approaches that seek to strengthen the lattice.

The presence of functional groups (LacNAc) as subcomponents of glycans, combined with our current understanding of Golgi transport, may also explain the curious use of three different nucleotides to charge the array of sugars used as substrate donors for glycosylation. These sugar-nucleotides are transported into the Golgi via an anti-port mechanism where cytosolic sugar-nucleotides are

exchanged for Golgi nucleotides generated by the action of glycosyltransferase enzymes. Increased availability and use of UDP-GlcNAc in the *trans* Golgi would increase uridine-monophosphate (UMP) levels, thereby driving anti-port of UDP-Galactose into the *trans* Golgi to promote further extension with poly-LacNAc rather than capping by sialic acid. Were sialic acid also charged with UDP rather than cytidine-monophosphate (CMP), UDP-GlcNAc usage in the *trans* Golgi would drive import of UDP-Sialic acid and capping equally with UDP-Galactose and extension.

## Materials and methods

### Galectin-3 binding
Recombinant mouse galectin-3 (R&D Systems, Minneapolis, MN) was labeled using an Alexa Fluor 488 protein labeling kit (ThermoFisher Scientific, Waltham, MA). Cells were stained for flow cytometry using 3 µg labeled galectin-3 per test. Staining was carried out for 30 min at room temperature followed by one wash and fixation with paraformaldehyde. For the lactose control samples, 50 mM lactose was included in all steps.

### PNGase F treatment of live cells
Purified mouse T cells were washed twice in HBSS and resuspended in 100 ul of G7 reaction buffer. 2500 units of glycerol free PNGase F (New England Biolabs, Ipswich, MA) was added to the cells and the reaction was allowed to proceed for up to four hours at 37 degrees Celsius. Cells were then washed with HBSS and immediately stained for flow cytometry.

### MALDI-TOF mass spectrometry
Control and SW treated Jurkat T cells were homogenized by sonication in 25 mM Tris, 150 mM NaCl, 5 mM EDTA, and 1% CHAPS, pH 7.4, dialyzed, reduced, and carboxymethylated, digested with trypsin to generate glycopeptides and treated with PNGase F (Roche, Basel, Switzerland) to release N-glycans. N-glycans were digested by endo-$\beta$-galactosidase (AMS Biotechnology) or Sialidase A (Prozyme, Hayward, CA) prior to permethylation and mass spectrometry analysis.

### Cell lines
Human Jurkat cell line E6-1 and its derivative cell lines with deficiency in TCR$\beta$ (J.RT3-T3.5), and Lck (J.CaM1.6) were purchased from ATCC. Two separate lots of E6-1 Jurkat cells were purchase throughout the study and found to be equivalent in LEA increase in response to SW treatment. Cells were cultured, expanded and frozen down to create low passage stocks. These were subsequently thawed and used for experiments, limiting usage to a maximum of ten passages. They were grown in RPMI 1640 medium with 10% fetal bovine serum, 2 mM L-glutamine, 100 U/ml penicillin/streptomycin and 2 µM 2-mercaptoethanol. CFPAC-1, CHO, Lec1, Raji, RPMI 8226, RS4; 11 and Kasumi-1 cell lines were also purchased from the ATCC. BJ, HT-29 and HEK-293 cells were a kind gift from Bogi Andersen. MCF-7 and Hela cells were a kind gift from Marian Waterman. Mouse postnatal day 1 neural stem cells were a kind gift from Thomas Lane. HUVEC cells were a kind gift from Christopher Hughes, and K562 cells were a kind gift from David Fruman. No routine mycoplasma testing or identity authentication was performed.

### Reagents, mice and flow cytometry
Isolated human CD3$^+$ T cells purified by negative selection (RosetteSep, StemCell Technologies, Vancouver, Canada) were stimulated with plate-bound anti-CD3 (OKT3, eBioscience, San Diego, CA). Procedures with human subjects were approved by the Institutional Review Board of the University of California, Irvine. *Mgat2*$^{f/f}$ (006892), *Mgat1*$^{f/f}$ (006891) and *Lck-Cre* (003802) mice were obtained from Jackson Laboratory. *B3gnt2*$^{+/-}$ (11605) embryos were from the MMRRC and were rederived by the UC Irvine Transgenic Mouse Facility. Inter-breeding generated all other mice. Mice were selected randomly for experiments and approved by the Institutional Animal Care and Use Committee of the University of California, Irvine. Human cells were stained with anti-CD4, anti-CD69 (eBioscience) and/or L-PHA-FITC (*Phaseolus Vulgaris* Leukoagglutinin, Vector Labs, Burlingame, CA). Mouse cells were stained with anti-CD4 (RM 4–5), anti-CD8 (53–6.7), and anti-CD69 (H1.2F3) (eBioscience). Proliferation was assessed by 5, 6-carboxyfluorescein diacetate

succinimidyl ester (CFSE; ThermoFisher Scientific) at 1 uM in PBS (20 min., 4°C) and stimulated with plate-bound anti-CD3e. Cells were cultured in RPMI-1640 media supplemented with 10% fetal bovine serum, 2 µM L-glutamine, 100 U/ml penicillin/streptomycin and 2 µM 2-mercaptoethanol. Where indicated, 40 mM GlcNAc (Ultimate Glucosamine, Wellesley Therapeutics Inc., Toronto, Canada) and 10 mM uridine (Sigma-Aldrich, St Louis, MO) were added to cells in culture at time 0. For flow cytometric analysis of glycan expression, cells were stained with 2 µg/ml L-PHA-FITC, ConA-FITC, LEA-FITC or biotinylated versions of these lectins followed by DyLight649 labeled streptavidin (Vector Labs). Staining was carried out as previously described (*Lee et al., 2007*; *Grigorian et al., 2007*). Flow cytometry experiments were performed with the BD FACSCalibur, LSR II, or Attune Acoustic Focusing Cytometer. Data analysis was performed using FlowJo software.

## RNA isolation and microarray

CD4[+] T cells isolated from *Mgat2*[f/f] mice and L-PHA[-]CD4[+] T cells isolated from *Mgat2*[f/f]::*Lck-Cre*[+] mice were used for analysis. The RNeasy mini kit (Qiagen, Valencia, CA) was used for RNA extraction. Gene expression was assessed using the Affymetrix Mouse Gene 1.0 ST arrays in triplicate. Array data were quantified with Expression Console version 1.1 software (Affymetrix, Santa Clara, CA) using the PLIER Algorithm default values. Expression values were then filtered as present/absent at expression 100. The Cyber-T web server was used for data analysis and to compare samples.

## Enzymatic assays

B3GNT enzyme activity was measured using a glycosyltransferase activity kit (R&D systems). Five million Jurkat cells grown with or without SW were washed three times with Tris buffered saline and lysed with 300 µl of lysis buffer (10 mM Tris pH 7.5, 2 mM MnCl₂, 4 mM CaCl₂, 0.5% Triton-X, with protease inhibitors). The lysate was cleared of insoluble material by centrifugation and dialyzed overnight in an identical solution to remove cellular phosphate. Enzyme activity was measured following the instructions of the kit. Briefly, 25 µl of lysate was mixed with 25 µl of reaction mixture to give a final concentration of 6 mM N-acetyllactosamine, 5 mM UDP-GlcNAc, and 6 ng/µl coupling phosphatase. The reaction was allowed to proceed for two hours at 37°C, followed by visualization of released phosphate by a malachite green reagent as indicated in the kit. Absorbance at 620 nm was determined using a plate reader and converted using a phosphate standard run in parallel. Background was determined and subtracted by parallel reactions that lacked the specific acceptor N-acetyllactosamine but contained UDP-GlcNAc and coupling phosphatase.

## Experimental autoimmune encephalomyelitis

EAE was induced by subcutaneous immunization of randomly selected female mice at days 0 with 100 µg of bovine MOG35-55 (AnaSpec, Fremont, CA) emulsified in Complete Freund's Adjuvant containing 4 mg/ml heat-inactivated *Mycobacterium tuberculosis* (H37RA; Difco). On days 0 and 2, mice received 150 ng of pertussis toxin (List Biological Laboratories, Campbell, CA) by intraperitoneal injection. Mice were examined daily for clinical signs of EAE over the next 30 days with observer blinded to treatment conditions. Mice were scored daily as follows: 0, no disease; 1, loss of tail tone; 2, hindlimb weakness; 3, hindlimb paralysis; 4, forelimb weakness or paralysis and hindlimb paralysis; 5, moribund or dead. Kifunensine was given orally via the drinking water at 0.2 mg/ml for 8 days starting 3 days before immunization. All mice were housed with 12 hr light/dark cycles.

## Spectrophotometric measurement of UDP-GlcNAc

UDP-GlcNAc was measured spectrophotometrically as previously described (*Barthel et al., 2011*). Briefly, Jurkat cells were grown in various conditions, washed thoroughly with PBS and pelleted at $50 \times 10^6$ cells per 1.5 ml tube. Cells were lysed by addition of 200 µl of chloroform/water (1:1), vortexed for 2 min and centrifuged at 15,000 × g for 20 min at 4°C. The aqueous phase was transferred to a fresh tube and 10 µl of 1 N HCl was added to hydrolyze UDP-GlcNAc to GlcNAc. After heating for 20 min at 80°C, the sample was neutralized with 10 µl 1 N KOH. Next, 50 µl of 200 mM potassium tetraborate (Sigma-Aldrich) was added, and the sample was heated at 80°C for 25 min and then cooled on ice for 5 min. 150 µl of Ehrlich's reagent (Sigma-Aldrich) (diluted 1:2 in acetic acid)

was added to the sample, mixed, and incubated for 20 min at 37°C. The sample was centrifuged at 15,000 × g for 20 min, 200 µl were added to a 96-well plate, and the absorbance was measured at 595 nm. Absorbance was converted to UDP-GlcNAc concentration by comparing to a GlcNAc standard curve run simultaneously.

## UDP-GlcNAc measurement by LC-MS/MS

Jurkat and mouse T cells were treated as described in the text, then harvested, washed twice with ice-cold 1x PBS and counted. After normalizing for cell number, cells were pelleted in 1.5 ml Eppendorf tubes by centrifuging at 350 g for 7 min at 4°C. Metabolites were extracted from the pellets by the addition of 1 ml of ice-cold solution of 40% acetonitrile, 40% methanol, and 20% water. The tubes were vortexed for 1 hr at 4°C and spun down at 14000 rpm for 10 min at 4°C (Eppendorf, Hamburg, Germany). The supernatant was transferred to fresh tubes and evaporated to dryness in an Eppendorf Vacufuge at 30°C (Eppendorf). The dry samples were stored at -80°C until analysis at which point they were reconstituted in 100 µl of LC-MS grade water and centrifuged again at 14000 rpm for 10 min at 4°C. The supernatants were carefully transferred to fresh tubes to be analyzed by mass spectrometry. The instrument used was a Waters Quattro Premier XE LC-MS/MS (Manchester, UK) equipped with ultra-performance liquid chromatography (UPLC) and a refrigerated autosampler. The separation was performed on BEH C18 reversed phase column (Waters, Manchester, UK) with a mobile phase A containing LC-MS grade water (JT Baker) modified with 0.2% Acetic Acid, 2% Acetonitrile and 5 mM Ammonium Acetate while mobile B containing LC-MS grade Acetonitrile (JT Baker) modified with 0.2% Acetic Acid and 5 mM ammonium Acetate. The gradient was performed with a 1 min ramp from 2% mobile B to 90%, which was then kept at 90% for 1 more minute and ramped back to 2% for column equilibration. The mass spectrometry measurement was performed in negative mode where the parent ion and fragment ion were measured simultaneously to be used for quantitation of the metabolites. A standard curve was generated using UDP-GlcNAc (Sigma-Aldrich), which was used for the quantitation of the metabolites produced by the cells.

## Cellular subfractionation and Golgi immunoisolation

Jurkat cells were grown with or without kifunensine for four days, or until cultures reached a density between 1.5–2 x 10^6 cells/ml. Cells used for Golgi compartment disruption were treated with 1000x Brefeldin A (eBioscience 00-4506-51) for up to 30 min. Cells were washed with HEPES buffered saline (145 mM NaCl, 100 µM HEPES, pH 7.4) and incubated in an ice-cold hypotonic solution (30 mM KCl, 3 mM MgOAc, 2 mM DTT, 20 mM HEPES, pH 7.5) for 5 min on ice. The volume of the hypotonic solution used was 2.5 times the volume of the cell pellet. The cell solution was then passed through a 26.5 gauge syringe five times before a 1/10 volume of ice-cold hypertonic solution (375 mM KCl, 22.5 mM MgOAc, 1 mM DTT, 220 mM HEPES, pH 7.5) was added. The solution was centrifuged twice in succession at 1000x g for 5 min each to pellet cells and nuclei. The post-nuclear supernatant (PNS) was carefully transferred to a new tube and processed further. Ultracentrifugation of the PNS at 100,000 g for 30 min yielded a vesicular fraction (the pellet) and a cytosolic fraction (the supernatant). For immunoisolation experiments, PNS was incubated with anti-TNG46 antibody (Sigma-Aldrich T7576-200UL) previously conjugated to Protein G Dynabeads (ThermoFisher Scientific) for 30 min at 4°C with tumbling. Supernatant was removed, and immunoisolated Golgi was washed with HEPES buffered saline before processing for mass spectrometry or immunoblot analysis.

## Plasmids, transfection and microscopy

DNA encoding mVenus and human *B3GNT2* (HA tagged at the C terminus) were codon-optimized for human expression, synthesized, and cloned into the pmax Cloning vector (Lonza, Basel, Switzerland). Myc-DDK tagged human *SLC35A3* (RC215108), *SLC35B4* (RC203263), and *SLC35D2* (RC218472) cDNA clones were purchased from Origene (Rockville, MD) in pCMV6 expression vectors. Cells were transfected with each construct individually (for microscopy) or in combination with mVenus to label transfected cells (for flow cytometry). 10^6 cells were transfected with 4–8 µg of DNA using an Amaxa Nucleofector IIb set to program G-10. Cells were immediately transferred to pre-equilibrated media in 6 well plates and placed in the tissue culture incubator. After 5 hr, media was changed and pharmacological treatments were started. For localization

studies, cells were transferred to poly-L-lysine coated chamber slides (Falcon) after 24–48 hr of treatment. Cells were incubated in the chamber slides for 15 min to allow attachment to the slides and then fixed for 1 hr with 4% paraformaldehyde in PBS. Standard immunocytochemical staining was performed. Permeabilization was achieved with 0.1% saponin (Sigma-Aldrich). Primary antibodies used were to GM130 (Clone 35, BD Biosciences, Franklin Lakes, NJ), TGN46 (AbD Serotec, Hercules, CA), Mannosidase II (a kind gift from Dr. Christine Suetterlin), HA (ab9110, Abcam, Cambridge, UK), and DDK (Clone 4C5, Origene). Slides were stained with the appropriate secondary antibodies conjugated to Alexa Fluor (AF) 488, 555 and 647 and mounted with Prolong Gold Antifade reagent with DAPI (ThermoFisher Scientific). GM130 and TGN46 were always visualized with AF 488 and 647 respectively, while the protein of interest was visualized with AF 555. Slides were imaged on a Zeiss LSM 780 confocal microscope using a Zeiss plan-apochromat 100x oil objective with numerical aperture of 1.4. Localization of the proteins of interest was determined relative to GM130 and TGN46 using a line-scan method essentially identical to a previously published report (*Dejgaard et al., 2007*). Briefly, 30–50 transfected cells were imaged per experimental condition. Files were coded to blind the analyzer to treatment and protein of interest. Images were imported into the Zeiss Zen software (blue edition) with the AF 555 channel disabled. Line-scans (8–10 pixels wide) were drawn perpendicular to the long axis on areas of the Golgi showing maximal separation of GM130 and TGN46. Line-scans were used if they resulted in a single clearly discernable peak in all three channels. The relative distance of the peak intensity of the protein of interest compared to the peak intensities of GM130 and TGN46 was then used to determine intra-Golgi localization.

## Immunoblotting

Immunoblotting was performed as described (*Demetriou et al., 2001*; *Lee et al., 2007*; *Grigorian et al., 2007*). Antibodies against GM130 (610822), GS27 (611034), GS28 (611184), and STX6 (610635) were from BD Biosciences. The TGN46 antibodies used for immunoisolation (T7576) and immunoblotting (SAB4200235) were from Sigma, and the antibody against B3GNT2 was from Origene (TA505283).

## Statistical analysis

Statistics were calculated with Prism software (GraphPad). *P* values were from two- or one-tailed unpaired *t*-tests (with Welch's correction). The Bonferroni correction was applied to all multiple comparisons. No statistical method was used to predetermine sample size. The experiments were not randomized.

## Acknowledgements

Research was supported by the National Institute of Allergy and Infectious Diseases (R01AI053331, R01AI108917) and National Center for Complementary and Integrative Health (R01AT007452) to MD. HM was supported by National Heart, Lung, and Blood Institute of the National Institutes of Health under award number F30HL108451. Mass spectroscopy work was supported by the Biotechnology and Biological Sciences Research Council grant BB/K016164/1 (to AD and SMH). AD is a Wellcome Trust Senior Investigator. We thank Jesse Rodriguez for artistic assistance and Adeela Syed, John Greaves, Beniam Berhane and the UCI Genomics High-Throughput Facility for technical assistance.

## Additional information

### Funding

| Funder | Grant reference number | Author |
| --- | --- | --- |
| National Heart, Lung, and Blood Institute | F30HL108451 | Haik Mkhikian |
| Biotechnology and Biological Sciences Research Council | BB/K016164/1 | Stuart M Haslam<br>Anne Dell |

| Wellcome Trust | | Anne Dell |
| National Institute of Allergy and Infectious Diseases | R01AI053331 | Michael Demetriou |
| National Center for Complementary and Integrative Health | R01AT007452 | Michael Demetriou |
| National Institute of Allergy and Infectious Diseases | R01AI108917 | Michael Demetriou |

The funders had no role in study design, data collection and interpretation, or the decision to submit the work for publication.

#### Author contributions

HM, Conception and design, Acquisition of data, Analysis and interpretation of data, Drafting or revising the article; C-LM, RWZ, KK, GW, PK, Acquisition of data, Analysis and interpretation of data; SMH, AD, Acquisition of data, Analysis and interpretation of data, Drafting or revising the article; MD, Conception and design, Analysis and interpretation of data, Drafting or revising the article

#### Author ORCIDs

Michael Demetriou, http://orcid.org/0000-0001-8547-5774

#### Ethics

Human subjects: Informed consent was obtained from human subjects to obtain peripheral blood for isolation of T cells and that resulting publications and/or presentations will not contain identifiable information. This was approved by the University of California Irvine Institutional Review board (HS#2001-2075).

Animal experimentation: This study was performed in strict accordance with the recommendations in the Guide for the Care and Use of Laboratory Animals of the National Institutes of Health. All of the animals were handled according to approved institutional animal care and use committee (IACUC) protocols (#2001-2305) of the University of California, Irvine.

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
