## [Decision Letter]

Thank you for submitting your article "Golgi proofreading homeostatically controls cell growth and differentiation" for consideration by *eLife*. Your article has been favorably evaluated by Vivek Malhotra (Senior editor) and four reviewers, one of whom, Benjamin Glick, is a member of our Board of Reviewing Editors. One of the reviewers has agreed to reveal his identity: Hudson Freeze.

The reviewers have discussed the reviews with one another and the Reviewing Editor has drafted this decision to help you prepare a revised submission.

Summary:

The four reviewers are in agreement that your manuscript is interesting, thorough, and of high quality and potentially of broad interest.

Our major concern is that there is no evidence that the phenomena you have described reflect a physiologically relevant homeostatic proofreading mechanism. This speculative aspect of the study needs to be toned down.

The main insight is bio-equivalence of the different LacNAc structures. This conclusion should be the main emphasis, and the title should be revised accordingly. One reviewer noted that "bio-equivalent glycans can be seen in many places – for instance man-6-P on various structures of high mannose glycans on lysosomal enzymes, or the MECA79 antigen structures as extended Core 1 structures rather than exclusively Core2." Therefore, your results should be placed in a more general context.

Essential revisions:

As indicated above, the emphasis of the paper needs to be adjusted to reflect the strong evidence for glycan bio-equivalence rather than the more speculative evidence for a homeostatic proofreading mechanism.

For the revision, please evaluate the detailed suggestions according to your best judgment, and make changes that you are confident will improve the paper.

Reviewer #1:

This manuscript by Haik Mkhikian et al. describes a mechanism by which the Golgi regulates the type of N-glycans it produces. With this mechanism, N-glycans are converted from highly branched to single-branched elongated LacNac structures. Both types of structures have high avidity for the secreted protein Galectin-3. This galectin forms a Galectin-glycoprotein lattice at the cell surface that regulates cell proliferation and differentiation of T-cells.

The paper proposes that upon perturbation of N-glycan branching by either genetic or chemical perturbation, cells react by increasing the amount of elongated LacNac structures through increasing the substrate required for LacNac, UDP-GlcNac, in the trans cisternae of the Golgi. GlcNac is usually present only in cis-medial Golgi, but lack of use by the branching enzymes is proposed to lead to its diffusion to the trans cisternae.

Overall, this manuscript and the data presented are of high quality and very exhaustive. The question tackled, the regulation of protein glycosylation, is not easy and in need of this type of analysis. The methods used by the authors are varied and cutting-edge, for instance the use of mass spec analysis for glycome analysis.

In terms of results, the increase in LacNac structures as detected by LEA staining is particularly striking and is reproducible across multiple systems.

However, I am not convinced by the notion of proof-reading for the following reasons:

The correction mechanism seems to be kicking in only after an extensive perturbation and is not completely correcting: the loss of Mgat5 or Mgat2 does not lead to compensation of Galectin biding to the wild-type levels. In fact, loss of Mgat5 does not induce any significant increase in LacNac branches. This is at odds with the authors model, since levels of UDP-GlcNac would be expected to increase in trans cisternae. In addition, this result suggests a very sloppy "proof-reading" system.

The perturbations used by the authors are artificial: genetic depletion of Mgat2 or treatment with swainsonine (SW). It is not clear what kind of naturally occurring problems are being mimicked by these treatments. Proof-reading implies that a system is sensing a defect and reacting to correct it. However, in this case, it is not the state of the lattice that is being sensed (or very mildly as per the authors own admission) but the level of use of sugar precursors (UDP-GlcNac) in the cis/medial Golgi. The compensatory mechanism is relatively indirect: other perturbations in the levels of UDP-GlcNac (for instance a change in the amount of cargo being traffic and glycosylated) would presumably lead to change in glycosylation that are unrelated to the status of the lattice.

Therefore, while the study is interesting and very well executed, at this stage, it is difficult to ascertain whether the phenomenon observed, the reality of which I do not doubt, is a compensatory mechanism occurring only in experimental conditions or whether it has a real biological function in a physiological context.

Reviewer #2:

The authors present a novel perspective for maintaining glycosylation homeostasis. They argue that "bio-equivalent" structures can substitute for certain functional glycans whose structure has been shown to be critical in determining cancer metastasis and autoimmunity. This lab and others have shown that the presence of poly-N-acetyllactosamine on the N-glycan branch created by an N-acetyl-Glucosaminyltransferse (GNT-V, encoded by MgatV) is essential for creation of lattices (or aggregate) composed of protein-bound glycans and multivalent lectins called galectins. This was a novel perspective when it was introduced 15 years ago. Evidence favoring it has grown in time and its functionality has been documented in deletion of GNT-V (MgatV) that leads to the loss of poly-N-acetyllactosamine on that branch, loss of galectin binding and dramatic decrease in tumor metastasis and increase in autoimmunity. Normally the lattice appears to prevent the inappropriate activation of the T-cell receptor. With the lattice gone, the TCR is able to recruit other cell surface molecules to mount T-cell response.

In this paper, the authors ask what would happen if two additional N-glycan branches were removed? Presumably, this would further reduce lectin binding and lattice formation and lead to more severe functional response. This did not occur. Instead they found a compensatory functional response in which the poly-N-acetyllactosamine repeats are added to another branch that normally would not contain them. To explain this, they propose a mechanism based on donor substrate localization to account for the compensation and float the hypothesis that generating bioequivalent structures may be a common way to maintain homeostasis in response to imbalances or insults. This is a novel perspective.

It is known that the terminal sugars of both N and O-linked glycans, especially repeating units, have a preferential site of addition, but it is generally not restricted to a single site.

In the Abstract, the word "transport" is used to refer to the movement of already transported substrates from the medial to the trans Golgi compartment. It is much more likely that it is simply diffusion from the medial to the trans Golgi. The existence of both vertical and horizontal inter-cisternal tubules has been known for some time. An important, but obscure and often overlooked paper 15 years ago (PMID:12376727) shows seven glycosyltransferases from different Golgi stacks and 3 nucleotide sugar transporters are functionally co-localized in a single continuous compartment that is accessible to any of the transported donors. They are freely diffusible within the Golgi and do not depend on any vesicular transport. This reference was not included, but must be acknowledged.

I think the authors adapt their results into a rationale for its existence. I hesitate to call it a proofreading mechanism to sustain homeostasis, but a much simpler explanation is that if a donor substrate is not used in a preferred location and it has access to a similar reaction in another accessible location, this is almost an inevitable outcome.

The study is thorough and very well done, using novel approaches and sophisticated techniques to make their arguments.

In Figure 2, the authors say there are no changes in Lec1 cells, yet it rates "**" significant differences. Need to reconcile those conflicting statements.

The MALDI results in Figure 2 as well as similar ones in the supplemental figures are impossible for a reader to quantitatively assemble and make conclusions. The increased ratio of 0.5 to 1.1 in the presence of SW may be true, but the authors need to show this via color-coding or another way to indicate which peaks are to be in the numerators and denominators. This is especially important when isobaric masses have different structures. It was very confusing.

The authors need to more precisely indicate which figure corresponds to which statement in the text because they are often confusing when multiple figures are indicated and statement refers to figures with many panels. For instance, the comment about a 3-4 fold increase in UDP-GlcNAc is enhanced SW induced poly-LacNAc extension in Figure 4 and Figure 4—figure supplement 1. Too much searching needed.

The statement in the Results section – "This suggests that cytosolic UDP-GlcNAc lacks direct access to the trans Golgi compartment" is not necessarily true. It may have been transported into a different compartment, but based on PMID:12376727, it likely has access to all compartments.

Likewise suggesting that the trans Golgi is deficient in UDP-GlcNAc, thereby limiting β3GnT activity and poly-LacNAc extension under steady state conditions is not necessarily true because the assumption relies on discreet non-interconnected compartments.

Reviewer #3:

In this manuscript the authors propose "that the cis to trans-Golgi organization permits a self-correcting mechanism that serves to preserve cellular homeostasis by maintaining galectin-glycoprotein interactions." And that "new poly-LacNAc extension by β3GnT maintains galectin binding and immune homeostasis. Poly-LacNAc extension is triggered by transport of unused UDP-GlcNAc from the medial to trans-Golgi via inter-cisternal tubules." This hypothesis is novel and the data presented in support of it are of high quality. The following points should be addressed in revision:

1) Some of the authors of this manuscript have previously published nice evidence of glycan elongation as a compensatory mechanism in glycoproteins from core 2 knockout mice. These and other results on this topic should be clearly described in the Introduction and relevant publications quoted.

2) The authors emphasize that a large drop in poly-LacNAc occurs in glycoproteins from Mgat2[-/-] compared to Mgat5[-/-] cells. However, based on the MS data, MGAT4 is operational in Mgat2[-/-] cells and thus the drop is merely from 3 to 2 branches. Going from 3 to 2 branches seems to be the threshold that stimulates the synthesis of LEA binding glycans.

3) The authors use mixed sugar symbols for MS spectra, the pathway diagram and Figure 7. It is strongly recommended that the symbols used on the MS spectra be used in all figures in the manuscript.

4) A key question that is not addressed is how much LacNAc is on each branch of N-glycans from WT vs. Mgat2 KO vs. Mgat5 KO T cells? The authors should note why they confined MS experiments to Jurkat cells. Given the large increase in LEA binding to Jurkat cells treated with SW, it is surprising that Figure 2 show a maximum of only 3 LacNAcs on 2 branches, but that peak is extremely small. By far the predominate species with a LacNAc extension has only 1 LacNAc (m/z 3102). This is not poly-LacNAc. The authors should discuss this disconnect.

5) The evidence that UDP-GlcNAc moves to the Trans Golgi when use in the medial Golgi decreases is largely indirect but reasonable, although many differences are small and there is a reliance on inhibitors that give modest effects and whose specificity is not investigated. The authors should discuss qualifications and potential caveats in the Discussion.

6) Bioequivalence is a fine concept but should be viewed as one function of surface glycans that is distinct from but complementary with biospecificity. It is not helpful to so dogmatically put all eggs in a bioequivalence basket.

*Reviewer #4:*

The galectin-glycoprotein lattice is a dynamic extracellular structure important for receptor localization and function. This paper focuses on quality control of the N-acetyllactosamine (LacNAc) component, which binds galectin. The hypothesis is that cells employ a homeostatic regulatory mechanism to ensure a suitable level of LacNAc in their extracellular N-glycans. Results presented here suggest that the Golgi has a built-in self-correction mechanism in which a deficiency in synthesis of LacNAc branches by cis/medial-Golgi Mgat enzymes is compensated for by enhanced synthesis of linear poly-LacNAc chains by the β-1,3-N-acetylglucosaminyltransferase (B3GnT) enzyme in the trans-Golgi. This compensation is proposed to occur when reduced branching enzyme activity results in elevated levels of the UDP-GlcNAc substrate becoming available in the trans-Golgi for linear poly-LacNAc synthesis.

The conclusions of this paper have two broad implications. First, it seems that the branched and linear LacNAc components are functionally interchangeable with regard to galectin binding. If so, the "glycan code" may be much less complex than has been assumed. Second, therapeutic approaches that target the galectin-glycoprotein lattice can be made more effective by taking into account functional redundancy in the glycan linkages. Both of these insights could be quite significant, making the paper potentially suitable for *eLife*.

This paper is a dense read but is well written, logical, and rigorously controlled. Although I would defer to other referees for evaluating the immunology, the glycobiology and cell biology aspects strike me as largely persuasive, with the following caveats.

1) All of the experiments employ artificial manipulations such as gene knockouts and glycosylation inhibitors. Is there any justification for assuming that activity of the Mgat branching enzymes can be perturbed under conditions that prevail in an organism? In other words, is this a physiologically relevant phenomenon? I'm not sure if this question can be answered definitively, but it would be nice to see an explanation of how such a perturbation might occur.

2) Is it clear that B3GnT is normally limited by the levels of UDP-GlcNAc in the trans-Golgi? If so, why doesn't addition of GlcNAc promote poly-LacNAc formation in the absence of swainsonine? It seems unlikely that all of the excess UDP-GlcNAc generated by GlcNAc addition would be consumed by increased branching, so there should be more UDP-GlcNAc available for B3GnT.

3) While reading the paper, I wondered from the beginning why sialic acid capping would not block LacNAc extension. This issue was not mentioned until the very end of the Discussion. Is it possible that regulation of poly-LacNAc synthesis occurs at the level of sialic acid capping? If so, an entire dimension may be missing from the story. For example, returning to #2 above, perhaps B3GnT is actually limited not by UDP-ClcNAc levels but rather by sialic acid capping.

---

## [Author Response]

*[…] The main insight is bio-equivalence of the different LacNAc structures. This conclusion should be the main emphasis, and the title should be revised accordingly. One reviewer noted that "bio-equivalent glycans can be seen in many places – for instance man-6-P on various structures of high mannose glycans on lysosomal enzymes, or the MECA79 antigen structures as extended Core 1 structures rather than exclusively Core2." Therefore, your results should be placed in a more general context.*

*Essential revisions:*

*As indicated above, the emphasis of the paper needs to be adjusted to reflect the strong evidence for glycan bio-equivalence rather than the more speculative evidence for a homeostatic proofreading mechanism.*

For the revision, please evaluate the detailed suggestions according to your best judgment, and make changes that you are confident will improve the paper.

We would like to thank all of the editors and reviewers for their time and efforts in service of our manuscript. We cannot help but be pleased by the refreshingly well-reasoned responses and the expeditious overall process at *eLife*. We feel that the issues raised have led to valuable discussion and analysis among the authors and contributed considerably to an improved manuscript, which we submit along with this response.

The major points made by the reviewers are well taken. We have also previously debated the relative merits of emphasizing the bio-equivalence versus the proofreading aspects of the story. Given the fresh perspective of the expert panel, we are now also in agreement that the emphasis of the paper should be adjusted to highlight glycan bio-equivalence. We have re-worked the paper to better emphasize bio-equivalence. For example, we have removed the term “proof-reading” from the title and text, and describe this now as self-correction. We have also re-organized the paper to focus on bio-equivalence. Although we think it likely that a proof-reading like mechanism is truly at play, we recognize that the positive data directly supporting this hypothesis are limited. This is due in no small part to the fact that well established tools to analyze and track sugar nucleotides at sub-Golgi resolution currently do not exist. However, we are encouraged by the interest and discussion of our colleagues to pursue the issue further in future work.

Reviewer #1:

*[…] Overall, this manuscript and the data presented are of high quality and very exhaustive. The question tackled, the regulation of protein glycosylation, is not easy and in need of this type of analysis. The methods used by the authors are varied and cutting-edge, for instance the use of mass spec analysis for glycome analysis.*

*In terms of results, the increase in LacNac structures as detected by LEA staining is particularly striking and is reproducible across multiple systems.*

However, I am not convinced by the notion of proof-reading for the following reasons:

*The correction mechanism seems to be kicking in only after an extensive perturbation and is not completely correcting: the loss of Mgat5 or Mgat2 does not lead to compensation of Galectin biding to the wild-type levels. In fact, loss of Mgat5 does not induce any significant increase in LacNac branches. This is at odds with the authors model, since levels of UDP-GlcNac would be expected to increase in trans cisternae. In addition, this result suggests a very sloppy "proof-reading" system.*

These are great points. At the cell surface, mono- and bi-antennary glycans greatly outnumber tri- and tetra-antennary glycans, particularly in low branching cell types such as T cells. This is also apparent when we consider the relative affinity of the MGAT enzymes for UDP-GlcNAc. MGAT1 has the best affinity with Km ~0.04mM. The affinities decrease sequentially for MGAT2 (~0.9mM), MGAT4 (~4mM) and MGAT5 (~11mM). In other words, relative UDP-GlcNAc usage markedly decreases from MGAT1 to MGAT5. Therefore, according to our model, inhibition or loss of MGAT5 is expected to result in very little increase in UDP-GlcNAc availability to B3GNT2, as enzymes with much greater activity (i.e. MGAT1 and 2) will utilize the excess supply. In comparison, MGAT2 loss or inhibition would be expected to result in a much larger rise in UDP-GlcNAc levels in the trans Golgi. Note that B3GNT has a Km for UDP-GlcNAc that is comparable to MGAT2.

*The perturbations used by the authors are artificial: genetic depletion of Mgat2 or treatment with swainsonine (SW). It is not clear what kind of naturally occurring problems are being mimicked by these treatments.*

Our thinking is that this is a mechanism that is constantly at work to counter the uncertainty of a loosely directed and “template-less” synthetic process. In contrast, the synthesis of DNA, RNA, and protein, being template driven is guided by very specific instructions. To use the names of the processes literally, it is as if the assignment is to copy or transcribe or translate (word for word) an assigned text. Predicting the outcome is straightforward given the quality of the template and the competence of “the staff”. Glycan synthesis by comparison is akin to asking many more players to produce a finger painting under a specific rule set (you can only use certain fingers (monosaccarides) in specific orders and patterns). The “instruction” is specified by the enzymatic and chemical restrictions, but the resulting product cannot be predicted with anything near the exactness of template driven processes. In addition, the integrity of the genome is maintained in part by restricting DNA replication to a highly controlled context and defined period of time. Glycan biosynthesis by contrast must function in a variety of cellular states and be active virtually at all times. Nevertheless, the Golgi system is charged with maintaining the integrity of the glycome/lattice or risk disease. Just as you point out, the most obvious/likely situations in which such an assembly-line corrective mechanism might function is in response to changes in protein synthesis, metabolism, and cargo transit rates. It is a reasonable hypothesis that high protein transit rates through the medial Golgi would reduce branching efficiency by decreasing the time available for branching reactions. Under this scenario, having a back-up system that provides poly-LacNAc extension in the trans Golgi by capturing unused UDP-GlcNAc from the medial Golgi would be critical for the maintenance of the lattice at a minimal essential level. Even in an unstressed system, a second step in the assembly line would act to counteract the moment to moment variability and stochastic uncertainty of glycan synthesis. We believe that severe branching deficiency (induced by SW treatment or *Mgat2* deletion) uncovers this continual process which is difficult to appreciate at baseline due to technical limitations. This would be akin to DNA repair mechanisms, which are occurring all of the time but made more apparent in the context of super-physiological stress.

However, physiological relevance may also arise from environmental factors. For example, swainsonine is a natural compound found in plants (locoweed) and ingestion causes disease in livestock. Given the large number of genes involved in glycan biosynthesis, the number of naturally occurring substances which may interfere with these pathways is likely underappreciated. Corrective mechanisms that can buffer such environmental insults to glycan biosynthesis may play a physiological role in these situations.

Proof-reading implies that a system is sensing a defect and reacting to correct it. However, in this case, it is not the state of the lattice that is being sensed (or very mildly as per the authors own admission) but the level of use of sugar precursors (UDP-GlcNac) in the cis/medial Golgi. The compensatory mechanism is relatively indirect: other perturbations in the levels of UDP-GlcNac (for instance a change in the amount of cargo being traffic and glycosylated) would presumably lead to change in glycosylation that are unrelated to the status of the lattice.

We have removed the term “proof-reading” from the text and substituted it with more appropriate terminology such as “self-correction.”

Proofreading in its exact publishing meaning (we had to look it up) refers to comparing typeset with copy prior to mass production. In this sense it is restricted to situations when two texts are compared and assessed for typesetting errors, and therefore has strict analogy to template driven synthetic processes. We had used the term in its broader current usage, where it further applies to reading for grammar, syntax, spelling, flow, logic, internal consistency etc. (the strict term for which is copy editing). We agree with your assessment that what is being sensed is UDP-GlcNAc concentration, and thus reduced UDP-GlcNAc usage by the prior compartment. We are not suggesting that B3GNT2 is a proofreading enzyme specifically, but rather that the entire glycan biosynthetic system with its directional intra-Golgi organization contains a sequential assembly-line self-correcting capability. This function is important for preserving the galectin-glycoprotein lattice.

Therefore, while the study is interesting and very well executed, at this stage, it is difficult to ascertain whether the phenomenon observed, the reality of which I do not doubt, is a compensatory mechanism occurring only in experimental conditions or whether it has a real biological function in a physiological context.

We are excited that others share our interest in these issues. We are also intrigued by this very question of the physiologic role of this mechanism, and despite our arguments for its likely role, we acknowledge that direct evidence is currently lacking. We plan to actively explore these questions in the future.

Reviewer #2:

[…] In the Abstract, the word "transport" is used to refer to the movement of already transported substrates from the medial to the trans Golgi compartment. It is much more likely that it is simply diffusion from the medial to the trans Golgi. The existence of both vertical and horizontal inter-cisternal tubules has been known for some time. An important, but obscure and often overlooked paper 15 years ago (PMID:12376727) shows seven glycosyltransferases from different Golgi stacks and 3 nucleotide sugar transporters are functionally co-localized in a single continuous compartment that is accessible to any of the transported donors. They are freely diffusible within the Golgi and do not depend on any vesicular transport. This reference was not included, but must be acknowledged.

These are great points. We agree that UDP-GlcNAc likely diffuses or is carried forward by cisternal maturation. Thank you for pointing out the potential confusion with the word “transport.” The manuscript has been revised to replace “transport” with clearer wording. We also thank the reviewer for bringing to our attention this relevant reference, and we acknowledge that the Golgi is functionally interconnected and continuous via vertical and horizontal tubules. However, our data are consistent with a Golgi model that though continuous, is organizationally segmented such that enzymes can locally deplete substrate. A discussion of these points is now included in the revised manuscript.

I think the authors adapt their results into a rationale for its existence. I hesitate to call it a proofreading mechanism to sustain homeostasis, but a much simpler explanation is that if a donor substrate is not used in a preferred location and it has access to a similar reaction in another accessible location, this is almost an inevitable outcome.

This view assumes that the Golgi is a one-pot system, which our data argue against. Our view is that the organization of the Golgi (with its sub-compartments and specific set of particularly localized resident enzymes and transporters) has arisen due to evolutionary pressures that provide advantages over other arrangements. For example, our data indicate that the trans Golgi lacks direct access to UDP-GlcNAc via transporters and relies on diffusion from more proximal compartments, with the rate of diffusion limited by the utilization of UDP-GlcNAc in more proximal compartments. This is the result of the evolutionary driven placement of enzymes and transporters in different sub-compartments and therefore differs from a one-pot system.

*The study is thorough and very well done, using novel approaches and sophisticated techniques to make their arguments.*

In Figure 2, the authors say there are no changes in Lec1 cells, yet it rates "**" significant differences. Need to reconcile those conflicting statements.

Thank you for pointing this out. These asterisks were present in error and have been removed.

The MALDI results in Figure 2 as well as similar ones in the supplemental figures are impossible for a reader to quantitatively assemble and make conclusions. The increased ratio of 0.5 to 1.1 in the presence of SW may be true, but the authors need to show this via color-coding or another way to indicate which peaks are to be in the numerators and denominators. This is especially important when isobaric masses have different structures. It was very confusing.

Please see the response to Reviewer 1 above.

The authors need to more precisely indicate which figure corresponds to which statement in the text because they are often confusing when multiple figures are indicated and statement refers to figures with many panels. For instance, the comment about a 3-4 fold increase in UDP-GlcNAc is enhanced SW induced poly-LacNAc extension in Figure 4 and Figure 4—figure supplement 1. Too much searching needed.

This point is well taken. We have revised the manuscript to contain reference to specific lettered panels in the figure supplements where appropriate.

*The statement in the Results section – "This suggests that cytosolic UDP-GlcNAc lacks direct access to the trans Golgi compartment" is not necessarily true. It may have been transported into a different compartment, but based on PMID:12376727, it likely has access to all compartments.*

Likewise suggesting that the trans Golgi is deficient in UDP-GlcNAc, thereby limiting β3GnT activity and poly-LacNAc extension under steady state conditions is not necessarily true because the assumption relies on discreet non-interconnected compartments.

Again, we agree that the Golgi compartments are interconnected. However, based on our data, we suggest that the segmented organization of enzymes and transporters allows for local depletion of substrate prior to diffusion to other compartments. As we attempt to illustrate in our model in Figure 7, if UDP-GlcNAc is transported to the cis/medial Golgi, rapid use by the branching enzymes will limit diffusion to the later compartments despite intra-Golgi continuities.

Reviewer #3:

*In this manuscript the authors propose "that the cis to trans-Golgi organization permits a self-correcting mechanism that serves to preserve cellular homeostasis by maintaining galectin-glycoprotein interactions." And that "new poly-LacNAc extension by β3GnT maintains galectin binding and immune homeostasis. Poly-LacNAc extension is triggered by transport of unused UDP-GlcNAc from the medial to trans-Golgi via inter-cisternal tubules." This hypothesis is novel and the data presented in support of it are of high quality. The following points should be addressed in revision:*

1) Some of the authors of this manuscript have previously published nice evidence of glycan elongation as a compensatory mechanism in glycoproteins from core 2 knockout mice. These and other results on this topic should be clearly described in the Introduction and relevant publications quoted.

While some of our authors have published data on unusual structures induced in glycan deficient mice, the possibility that these represent a functional compensation mechanism were largely proposed in reviews by others. Importantly, to our knowledge no one has previously directly proven that the induced structures were bio-equivalent and able to functionally compensate for the loss of other glycans. We now describe this issue in the Introduction.

2) The authors emphasize that a large drop in poly-LacNAc occurs in glycoproteins from Mgat2[-/-] compared to Mgat5[-/-] cells. However, based on the MS data, MGAT4 is operational in Mgat2[-/-] cells and thus the drop is merely from 3 to 2 branches. Going from 3 to 2 branches seems to be the threshold that stimulates the synthesis of LEA binding glycans.

These are important points. We state that there is a significant drop in LacNAc branching/content not poly-LacNAc (we assume this is a typo in your comment) between *Mgat5* and *Mgat2* deficient cells. Also, as you point out below, the MS data are from SW treated Jurkat cells not *Mgat2* deficient cells. Thus, it is possible that MGAT4 is not active in *Mgat2* deficient T cells. You are correct that the MGAT4 branch is present in the SW treated Jurkat cells. As noted above in the comments to reviewer 1, MGAT2 has a higher affinity for UDP-GlcNAc than MGAT4. Based on this we would predict that loss of MGAT2 activity or inhibition by SW (despite an operational MGAT4) may result in greater UDP-GlcNAc availability for poly-LacNAc extension than loss of MGAT4/MGAT5, despite both scenarios resulting in a maximum of two branches. In other words, going from 3 to 2 branches by combined loss of MGAT4/MGAT5 may not be equivalent to MGAT2 loss or SW treatment. Our model would suggest that UDP-GlcNAc usage by the branching enzymes rather than the number of branches maximally produced dictates extent of poly-LacNAc compensation. This is a subtle but important distinction.

3) The authors use mixed sugar symbols for MS spectra, the pathway diagram and Figure 7. It is strongly recommended that the symbols used on the MS spectra be used in all figures in the manuscript.

This point is well taken. The discordant symbols have been replaced with a uniform set.

4) A key question that is not addressed is how much LacNAc is on each branch of N-glycans from WT vs. Mgat2 KO vs. Mgat5 KO T cells? The authors should note why they confined MS experiments to Jurkat cells. Given the large increase in LEA binding to Jurkat cells treated with SW, it is surprising that Figure 2 show a maximum of only 3 LacNAcs on 2 branches, but that peak is extremely small. By far the predominate species with a LacNAc extension has only 1 LacNAc (m/z 3102). This is not poly-LacNAc. The authors should discuss this disconnect.

These are excellent points. Indeed, the *Mgat2* deficient cells exhibited far greater fold LEA increase (~100 fold) than the Jurkat cells (~20 fold). We therefore made several attempts to perform MS experiments on Mgat2 KO vs. WT T-cells. However, the analyses revealed that we were unable to isolate enough biological material to generate meaningful spectra (we only detected high mannose structures), despite pooling cells from many mice. This was due in part to the fact that we are dealing with a T cell specific knockout, which is inefficiently deleted, and mouse T cells yield relatively low amounts of glycan. We therefore resorted to the SW treated Jurkat cells as a reasonable alternative. As far as the extent to which poly-LacNAc is represented on the spectra, these have to be interpreted cautiously because they are not quantitative, particularly for large poly-LacNAc structures. For that reason we performed the endo-β-galactosidase digestion analysis, which revealed increased poly-LacNAc in the Jurkat sample.

5) The evidence that UDP-GlcNAc moves to the Trans Golgi when use in the medial Golgi decreases is largely indirect but reasonable, although many differences are small and there is a reliance on inhibitors that give modest effects and whose specificity is not investigated. The authors should discuss qualifications and potential caveats in the Discussion.

Some of the differences noted are indeed small. We suspect that this is because we are achieving mild relative enrichment of the late Golgi compartments, rather than isolating a pure fraction. This dilutes the true difference. As noted above, we have toned down our discussion on this subject by focusing the paper more on bio-equivalence than proof-reading.

6) Bioequivalence is a fine concept but should be viewed as one function of surface glycans that is distinct from but complementary with biospecificity. It is not helpful to so dogmatically put all eggs in a bioequivalence basket.

We agree about bio-specificity; however galectin bio-specificity is for LacNAc, rather than the entire glycan structure. The overall glycan structure sets the total LacNAc content, thereby largely determining avidity of binding without affecting affinity of binding (assuming other modifications that may impact binding such as fucose and sialic acid are held constant). Thus, we are not arguing that there is a change in bio-specificity of galectin-glycan interactions, just a recognition that the number of binding sites in the overall glycan (i.e. LacNAc content) is a major driver of bio-equivalence in a manner that does not affect bio-specificity.

Reviewer #4:

*[…] This paper is a dense read but is well written, logical, and rigorously controlled. Although I would defer to other referees for evaluating the immunology, the glycobiology and cell biology aspects strike me as largely persuasive, with the following caveats.*

1) All of the experiments employ artificial manipulations such as gene knockouts and glycosylation inhibitors. Is there any justification for assuming that activity of the Mgat branching enzymes can be perturbed under conditions that prevail in an organism? In other words, is this a physiologically relevant phenomenon? I'm not sure if this question can be answered definitively, but it would be nice to see an explanation of how such a perturbation might occur.

Please see above comment to reviewer 1.

2) Is it clear that B3GnT is normally limited by the levels of UDP-GlcNAc in the trans-Golgi? If so, why doesn't addition of GlcNAc promote poly-LacNAc formation in the absence of swainsonine? It seems unlikely that all of the excess UDP-GlcNAc generated by GlcNAc addition would be consumed by increased branching, so there should be more UDP-GlcNAc available for B3GnT.

GlcNAc treatment markedly increases total cellular UDP-GlcNAc and branching without substantial changes in LEA binding in the absence of swainsonine. Coupled with our data indicating that UDP-GlcNAc transporters are absent from the trans Golgi, we interpret this to mean that excess UDP-GlcNAc generated by GlcNAc addition is indeed consumed by the medial Golgi branching enzymes. In the presence of SW, GlcNAc treatment does substantially increase LEA binding, implying that UDP-GlcNAc is limiting for poly-LacNAc production but only becomes available to B3GNT when the branching enzymes are inhibited. It should be noted that an increase in total cellular UDP-GlcNAc may not necessarily result in one for one increase in Golgi UDP-GlcNAc. The apparent Km of the transporters is measured at 1-10 uM, which is lower than the cytosolic UDP-GlcNAc concentration. Therefore it is likely that increased cytosolic UDP-GlcNAc levels are not fully translated to the Golgi levels.

*3) While reading the paper, I wondered from the beginning why sialic acid capping would not block LacNAc extension. This issue was not mentioned until the very end of the Discussion. Is it possible that regulation of poly-LacNAc synthesis occurs at the level of sialic acid capping? If so, an entire dimension may be missing from the story. For example, returning to #2 above, perhaps B3GnT is actually limited not by UDP-ClcNAc levels but rather by sialic acid capping.*

Another excellent point. Sialic acid capping could be limiting poly-LacNAc length. However, we did not see significant changes in sialyltransferase transcript levels in our microarray experiment. Moreover, in our mass spec data there appears to be more sialylation of the poly-LacNAc antennae in the Swainsonine treated sample (compare m/z 722 and 1084 in Figure 2), arguing against reduced sialic acid capping activity resulting from SW treatment.